# CASE-GUIDED SEQUENTIAL ASSAY PLANNING IN DRUG DISCOVERY

## ABSTRACT

Optimally sequencing experimental assays in drug discovery is a high-stakes planning problem under severe uncertainty and resource constraints. A primary obstacle for standard reinforcement learning (RL) is the absence of an explicit environment simulator or transition data $(s, a, s')$; planning must rely solely on a static database of historical outcomes. We introduce the Implicit Bayesian Markov Decision Process (IBMDP), a model-based RL framework designed for such simulator-free settings. IBMDP constructs a case-guided implicit model of transition dynamics by forming a nonparametric belief distribution using similar historical outcomes. This mechanism enables Bayesian belief updating as evidence accumulates and employs ensemble MCTS planning to generate stable policies that balance information gain toward desired outcomes with resource efficiency. We validate IBMDP through comprehensive experiments. On a real-world central nervous system (CNS) drug discovery task, IBMDP reduced resource consumption by up to 92% compared to established heuristics while maintaining decision confidence. To rigorously assess decision quality, we also benchmarked IBMDP in a synthetic environment with a computable optimal policy. Our framework achieves significantly higher alignment with this optimal policy than a deterministic value iteration alternative that uses the same similarity-based model, demonstrating the superiority of our ensemble planner. IBMDP offers a practical solution for sequential experimental design in data-rich but simulator-poor domains.

## 1 INTRODUCTION

To discover new drugs, scientists make sequential decisions to conduct multiple assays, often constrained by limited time, budget, and materials. The process typically begins with sparse evidence from historical assay outcomes on past compounds. Executing an assay for a new drug candidate compound yields an observation of the assay consuming monetary and time resources, each probing a distinct facet of developability of the compound (e.g., potency, ADME, safety). For example, an *in vitro* assay may be cheap and fast but only weakly informative downstream, whereas an *in vivo* assay is slower and more expensive yet more decisive for Go/No-Go decisions. Under tight budget and schedule constraints, the central question is whether to run another assay or stop now. Ideally, each chosen assay reduces posterior uncertainty while increasing the likelihood that the compound satisfies predefined developability criteria. This is a planning problem under uncertainty, further complicated by the absence of transition tuples $(s, a, s')$—only historical assay outcomes from past compounds are available. In practice, rule-based playbooks and expert heuristics are often risk-averse or myopic, leading to inefficient use of constrained resources and suboptimal portfolio outcomes.

To address these challenges, we propose the Implicit Bayesian Markov Decision Process (IBMDP), a reinforcement learning (RL) framework for case-guided sequential assay planning that uses assay outcomes of historical compounds to construct an implicit probabilistic model of information gain acquired from assays. At each step, IBMDP forms a categorical distribution over historical compound records using a variance-normalized distance kernel and samples plausible assay outcomes consistent with the current partial evidence, thereby updating the candidate's observed state. This implicit, nonparametric transition model emphasizes contexts most relevant to the candidate without requiring an explicit mechanistic simulator. Planning is performed with Monte Carlo Tree Search with Double Progressive Widening (MCTS-DPW), and we run an ensemble of MCTS planners to reduce variance from both stochastic sampling and tree search; majority voting across runs yields

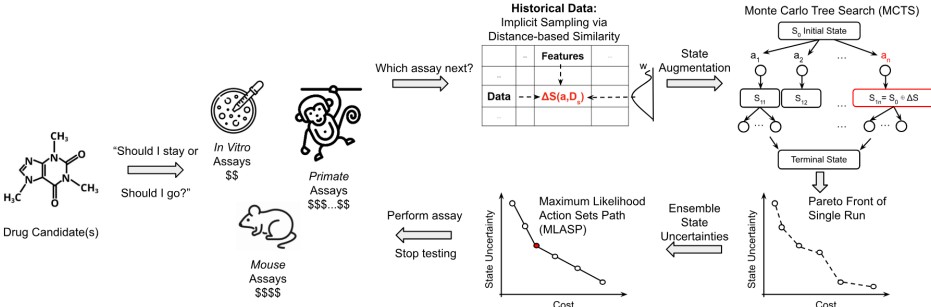

Figure 1: Sequential decisions in drug discovery through a data-driven, analog-guided simulator for planning, which maintains a Bayesian belief over the most relevant historical compound analogs.

a Maximum-Likelihood Action-Sets Path (MLASP) that is stable across uncertainty levels. When simulating possible courses of actions during search, IBMDP takes into account the resulting reduction in uncertainty towards desirable states (e.g., high drug likeliness *in vivo*) and recommends the next assay only when the uncertainty reduction reaches a sufficient magnitude towards the desirable states. From the Partially Observable MDP (POMDP) perspective, while standard methods maintain explicit probability distributions over hidden states and update them via Bayes' rule, IBMDP makes decisions by sampling from past experiences weighted by similarity to the current observed state (Appendix A). While IBMDP trades formal convergence guarantees for practical applicability in simulator-free settings, it provides empirically robust policies through ensemble MCTS planning (Appendix A.6).

**Contributions.** **(i) RL planning with evidence-adaptive dynamics:** Unlike traditional RL with fixed transition functions, IBMDP's implicit dynamics evolve as observations accumulate—the similarity-based belief continuously adapts, creating non-stationary but principled state transitions from static historical data. **(ii) Similarity-weighted Bayesian belief mechanism:** We transform historical outcomes into an adaptive generative model where transition probabilities dynamically shift based on accumulated evidence, enabling planning without explicit dynamics or $(s, a, s')$ trajectories. **(iii) Robust ensemble MCTS despite non-stationary dynamics:** Our ensemble approach with majority voting (MLASP) produces stable policies even with evolving transition models, optimally balancing information gain with resource efficiency.

## 2 PRELIMINARIES

**Compounds, Assays, and Historical Data** Let $\mathcal{A} = \{a_1, \ldots, a_M\}$ be the set of available assays and $X = \{x_1, \ldots, x_N\}$ be the set of historical compounds, each with a fixed molecular representation. For a historical compound $x_i$ and an assay $a_j$, the observed outcome is denoted by $y_{i,j}$. The complete historical dataset is represented as a set of tuples:

$$\mathcal{D} = \{(x_i, \mathbf{y}_i)\}_{i=1}^N,$$

where $\mathbf{y}_i = (y_{i,1}, \ldots, y_{i,M})$ is the vector of all assay outcomes for compound $x_i$. The new drug candidate compound for which we are planning is denoted $x_\star \equiv x_{N+1}$. We also have access to per-assay predictor models, such as Quantitative Structure-Activity Relationship (QSAR) models, which are functions $f_j : x \mapsto \hat{y}_j = f_j(x)$ that can be queried for the candidate $x_\star$ during the planning phase (Chen et al., 2024). For convenience, Appendix G collects all symbols used throughout the paper in the *Global Notation Reference*.

**Target Property** Let $g$ be the primary scalar target of interest, such as a definitive *in vivo* endpoint that determines a compound's success. The historical values for this target form the set $G = \{g_i\}_{i=1}^N$. In many applications, the target property may correspond to one of the available assays. That is, for some specific assay index $j \in \{1, \ldots, M\}$, we have $g_i \equiv y_{i,j}$ for all compounds. *Crucially*, to prevent data leakage, the set of target values $G$ is never used in the computation of similarity or distance metrics during planning. We define $I_g = \{i : g_i \text{ is available}\}$ as the set of indices for historical compounds where the target value has been measured.

**State and Actions** We formulate the assay planning problem for the candidate $x_\star$ as a finite-horizon MDP with a discount factor $\gamma \in [0, 1)$ and a maximum horizon $T$. At any decision step $t$, we maintain an index set of assays that have already been performed, $M_t \subseteq \{1, \ldots, M\}$, and the set of unmeasured assays, $U_t := \{1, \ldots, M\} \setminus M_t$. The process starts with an empty set of measured assays, $M_0 = \emptyset$.

The **state** at step $t$ summarizes all accumulated knowledge about the candidate compound: $s_t = \left(x_\star, \{y_{\star,j}\}_{j \in M_t}\right)$. The **action set** at step $t$, $\mathcal{A}_t$, consists of choosing a batch of up to $m$ currently unmeasured assays to perform, or deciding to stop the experiment. Formally: $\mathcal{A}_t = \mathcal{P}_{\leq m}(U_t) \cup \{\text{eox}\}$. Here, $\mathcal{P}_{\leq m}(U_t)$ is the set of all subsets of $U_t$ with size at most $m$, and 'eox' (end-of-experiment) is the terminal action. The parameter $m \leq M$ is a user-specified throughput limit that caps how many assays can be run in parallel at a single step. Executing an action $A_t \subseteq U_t$ reveals the outcomes $\{y_{\star,j}\}_{j \in A_t}$ and updates the measured and unmeasured sets for the next step, $t + 1$.

**Reward Function**  Each action incurs a cost based on the resources it consumes (e.g., time, materials, monetary expense). Let $c_j \in \mathbb{R}_{\geq 0}^q$ be the cost vector for an individual assay $a_j$. The cost for a batch action $A_t$ is the sum of the costs of the individual assays within it, i.e., $c(s_t, A_t) = \sum_{a_j \in A_t} c_j$. Let $\boldsymbol{\rho} \in \mathbb{R}_{\geq 0}^q$ be a user-defined vector of weights that specifies the trade-offs between different resources. The scalar step reward, $R(s_t, A_t)$, is defined as:

$$R(s_t, A_t) = \begin{cases} -\boldsymbol{\rho}^\mathsf{T} c(s_t, A_t), & \text{if } A_t \in \mathcal{A}_t \setminus \{\text{eox}\}, \\ 0, & \text{if } A_t = \text{eox}. \end{cases}$$

**Uncertainty and Goal-Likelihood Functionals**  To ensure resources are directed toward viable drug candidates, we define two key state-dependent scalar functions based on similarity weights $w_i(s_t)$ over historical records (to be formally defined in a later section). First, we renormalize the weights to consider only the historical compounds for which the target value $g$ is available:

$$\tilde{w}_i(s_t) = \frac{w_i(s_t)}{\sum_{\ell \in I_g} w_\ell(s_t)} \qquad \text{for } i \in I_g.$$

Note that when $|I_g| \ll N$, this renormalization may lead to variance underestimation as it restricts the effective sample size. This limitation is discussed in the experimental analysis. Using these normalized weights, we define:

1. **State-Uncertainty** ($H(s_t)$)**:** The weighted variance of the target property $g$ over the relevant historical data, which serves as a measure of uncertainty about the candidate's potential outcome.

$$H(s_t) = \sum_{i \in I_g} \tilde{w}_i(s_t) \left(g_i - \bar{g}(s_t)\right)^2, \quad \text{where} \quad \bar{g}(s_t) = \sum_{i \in I_g} \tilde{w}_i(s_t) g_i. \tag{1}$$

2. **Goal-Likelihood** ($L(s_t)$)**:** The weighted probability that the candidate's target property falls within a predefined desirable range $[g_{\min}, g_{\max}]$.

$$L(s_t) = \sum_{i \in I_g} \tilde{w}_i(s_t) \mathbf{1}[\, g_i \in [g_{\min}, g_{\max}]\,]. \tag{2}$$

Here, $\mathbf{1}[\cdot]$ denotes the indicator function, which returns 1 when its argument is true and 0 otherwise.

**Constrained Objective**  The optimal policy $\pi^*$ is one that maximizes the total expected reward, subject to constraints on terminal uncertainty and stepwise feasibility. Specifically, we aim to solve:

$$\pi^* \in \arg\max_\pi \mathbb{E}_\pi \left[\sum_{t=0}^T \gamma^t R\big(s_t, \pi(s_t)\big)\right]$$

$$\text{subject to} \quad \begin{cases} H(s_T) \leq \epsilon, & \epsilon \in [0, 1], \\ L(s_t) \geq \tau, & \forall t = 0, \dots, T-1, \end{cases} \tag{3}$$

where $\epsilon > 0$ is the maximum tolerable uncertainty at the terminal state $s_T$, and $\tau \in (0, 1)$ is the minimum acceptable goal-likelihood at every intermediate step. The feasibility constraint $L(s_t) \geq \tau$ ensures that the planning process remains on a trajectory toward a successful outcome, while the terminal constraint $H(s_T) \leq \epsilon$ guarantees that a decision is made with sufficient confidence.

## 3 IMPLICIT MODEL OF ENVIRONMENT DYNAMICS

The key challenge is updating the state transition $s_t \xrightarrow{A_t} s_{t+1}$—i.e., how the state evolves after executing a batch of assays—when no explicit simulator is available and only historical data $\mathcal{D}$ can

be leveraged to infer dynamics. We address this by constructing an implicit, generative model of the environment's dynamics. This model uses a similarity metric to dynamically re-weight historical compound profiles, forming a belief over plausible outcomes for the candidate compound $x_\star$. This avoids explicit parameterization of transition probabilities and implicitly propagates uncertainty by sampling from historical compound analogs most relevant to the current state of $x_\star$.

**Similarity Weight Computation**   The transition model is centered on a similarity weight, $w_i(s_t)$, assigned to each historical compound record $D_i = (x_i, \mathbf{y}_i) \in \mathcal{D}$. These weights quantify the relevance of each historical case to the current state, $s_t$. The weights are computed using a variance-normalized exponential kernel:

$$w_i(s_t) = \exp\left(-\lambda_w \cdot d(s_t, D_i)\right), \tag{4}$$

where $d(s_t, D_i)$ is a distance metric. The distance is computed over the set of all features known for the candidate $x_\star$ at step $t$, which we denote as the feature set $\mathcal{K}_t$. This set includes all initial QSAR predictions and the outcomes of all measured assays in $M_t$. The distance compares these known values for the candidate to the corresponding values for the historical compound $x_i$:

$$d(s_t, D_i) = \sum_{k \in \mathcal{K}_t} \lambda_k \cdot \frac{(\phi_k(s_t) - \phi_k(D_i))^2}{\sigma_k^2}, \tag{5}$$

Here, $\phi_k(\cdot)$ is an extractor function that returns the value of the $k$-th feature from a given state or historical record. For the candidate, $\phi_k(s_t)$ is either a QSAR prediction or a measured assay outcome $\{y_{\star,j}\}_{j \in M_t}$. For the historical compound, $\phi_k(D_i)$ is the corresponding recorded value. The term $\sigma_k^2$ is the empirical variance of feature $k$ across the historical dataset $\mathcal{D}$, computed as $\sigma_k^2 = \frac{1}{N}\sum_{i=1}^{N}(\phi_k(D_i) - \bar{\phi}_k)^2$ where $\bar{\phi}_k = \frac{1}{N}\sum_{i=1}^{N}\phi_k(D_i)$. The parameter $\lambda_k$ is a feature-specific weight, and $\lambda_w$ is a global temperature parameter. The variance normalization ensures a dimensionless comparison across features with different scales.

**Similarity-Based State Transition**   The transition from state $s_t$ to $s_{t+1}$ after executing an action (a batch of assays) $A_t \subseteq U_t$ is simulated through a weighted sampling process. First, a historical case is sampled from $\mathcal{D}$ with probability proportional to its similarity weight:

$$I \sim \text{Categorical}\left(\frac{w_1(s_t)}{Z}, \ldots, \frac{w_N(s_t)}{Z}\right), \quad \text{where } Z = \sum_{i=1}^{N} w_i(s_t).$$

Let the selected historical case be $D_I = (x_I, \mathbf{y}_I)$. The outcomes for the assays in the action batch $A_t$ are then "revealed" by taking the corresponding values from this sampled case:

$$\{y_{\star,j} := y_{I,j}\}_{j \in A_t}.$$

The new state $s_{t+1}$ is formed by augmenting the previous state with these newly generated outcomes. Formally, $M_{t+1} = M_t \cup A_t$, and the new state is:

$$s_{t+1} = \left(x_\star, \{y_{\star,j}\}_{j \in M_{t+1}}\right).$$

This generative process ensures that the simulated outcomes for the new assays are consistent with a plausible, historically observed compound profile, thereby preserving correlations between assays.

**Implicit Transition Modeling via Sampling**   The sampling mechanism described above defines an implicit transition probability distribution $P(s_{t+1}|s_t, A_t)$. This distribution is a mixture model where each component corresponds to one of the historical cases in $\mathcal{D}$. The probability of transitioning to a specific next state $s_{t+1}$ is the total weight of all historical cases that would produce that state:

$$P(s_{t+1}|s_t, A_t) = \sum_{i=1}^{N} \frac{w_i(s_t)}{Z} \cdot \mathbf{1}[s_{t+1} = s_t \oplus \{(a_j, y_{i,j})\}_{j \in A_t}], \tag{6}$$

where $\oplus$ denotes the state update operation that augments the current state by adding new assay outcomes to $M_t$ and updating the observed values $\{y_{\star,j}\}$, and $\mathbf{1}[\cdot]$ is the indicator function. This sampling-based approach approximates the true transition dynamics when the historical dataset $\mathcal{D}$ is sufficiently representative of the underlying system. The quality of the approximation depends on the coverage of $\mathcal{D}$, the appropriateness of the similarity metric, and the dataset size $N = |\mathcal{D}|$.

**Bayesian Weight Update**   After transitioning to the new state $s_{t+1}$, the similarity weights are re-evaluated to incorporate the new evidence. This recalculation is a direct and principled implementation of a Bayesian belief update. As we formally derive in Appendix A, our framework is

equivalent to a POMDP where the hidden state is a latent index over the historical cases in $\mathcal{D}$. The similarity weights $w_i(s_t)$ represent the posterior belief over these latent "prototypes," and the recalculation after observing new assay outcomes is equivalent to applying Bayes' rule. The new weights $\{w_i(s_{t+1})\}_{i=1}^N$ are computed using the same distance function as before, but now applied to the augmented state $s_{t+1}$:

$$w_i(s_{t+1}) = \exp\left(-\lambda_w \cdot d(s_{t+1}, D_i)\right). \tag{7}$$

This update mechanism shifts the model's belief toward historical cases that are most consistent with the expanded set of evidence for the candidate $x_\star$, allowing the planner to refine its predictions and subsequent decisions as more data is gathered.

# 4 Implicit Bayesian Markov Decision Process (IBMDP)

The Implicit Bayesian Markov Decision Process (IBMDP) is a planning framework designed to solve the constrained optimization problem defined in Equation equation 3. It integrates the implicit, case-based transition model with a powerful planning algorithm to find reward-maximizing sequences of assays. The core of the framework is a Monte Carlo Tree Search (MCTS) planner that navigates the decision space by simulating potential experimental paths using the generative model derived from historical data. To ensure the robustness of its recommendations, IBMDP employs an ensemble method, aggregating the results of multiple independent planning runs.

The overall workflow proceeds as follows: Historical Data $\mathcal{D}$ informs a Similarity Module, which computes weights $w_i(s_t)$ for the current state $s_t$. These weights drive the implicit transition model used by an MCTS-DPW planner. The planner generates a policy, and this process is repeated across an ensemble of runs. Finally, the policies are aggregated to construct a Maximum-Likelihood Action-Sets Path (MLASP), which constitutes the final recommended experimental plan. The detailed procedure is outlined in Algorithm 1.

---

**Algorithm 1** Ensemble IBMDP Algorithm

**Require:** Initial state $s_0 = (x_\star, M_0 = \emptyset)$, historical data $\mathcal{D}$, reward function $R(s, A)$, functionals $H(s), L(s)$, thresholds $\epsilon, \tau$, horizon $T$, iterations $n_{\text{itr}}$, ensemble size $N_e$.
**Ensure:** A Maximum-Likelihood Action-Sets Path (MLASP).
1: Initialize policy set $\Pi \leftarrow \emptyset$.
2: **for** $j = 1$ to $N_e$ **do**                          ▷ Ensemble loop
3:      Initialize MCTS tree $\mathcal{T}$ with root node $s_0$.
4:      **for** $i = 1$ to $n_{\text{itr}}$ **do**                    ▷ MCTS iterations
5:          **Selection:** Traverse $\mathcal{T}$ from $s_0$ using a tree policy (e.g., UCB1) to select a leaf node $s_{leaf}$.
6:          **Expansion:** If $s_{leaf}$ is not a terminal state ($H(s_{leaf}) > \epsilon$), choose an untried action $A \in \mathcal{A}_t(s_{leaf})$ and create a new child node $s_{new}$.
7:          **Simulation (Rollout):** From $s_{new}$, simulate a trajectory of states and actions using a reward-aware heuristic policy until a terminal state or horizon $T$ is reached.
8:             During rollout, for a transition $(s, A) \to s'$, the next state $s'$ is generated by the implicit model: sample $I \sim \text{Cat}(\{w_k(s)/Z\}_{k=1}^N)$ where $Z = \sum_{k=1}^N w_k(s)$ and set $s' = s \oplus \{(a_k, y_{I,k})\}_{k \in A}$.
9:             The total return $Q$ is the cumulative reward, with a large negative reward (e.g., $-10^6$) if $L(s) < \tau$ for any state in the trajectory.
10:         **Backpropagation:** Update the visit counts and value estimates for all nodes on the path from $s_{new}$ back to the root using the return $Q$.
11:      **end for**
12:      Extract the optimal policy $\pi_j^*$ from the final tree $\mathcal{T}$ by selecting the action with the highest value at each node.
13:      $\Pi \leftarrow \Pi \cup \{\pi_j^*\}$.
14: **end for**
15: Construct MLASP by aggregating policies in $\Pi$ via majority voting at each decision step.
16: **return** MLASP

---

IBMDP begins with the initial state $s_0$, which contains the candidate compound $x_\star$ and any preexisting QSAR predictions, with an empty set of measured assays ($M_0 = \emptyset$). The planner, MCTS with Double Progressive Widening (MCTS-DPW), is particularly well-suited for this problem due to its ability to handle large, combinatorial action spaces—in this case, the power set of unmeasured assays, $\mathcal{P}_{\leq m}(U_t)$.

During each simulation step within the MCTS algorithm, the planner must evaluate the consequence of taking an action $A_t$. It does this by invoking the implicit transition model from the previous section. A historical case $D_I$ is sampled based on the current similarity weights $w_i(s_t)$, and the

outcomes for the assays in $A_t$ are drawn from this case. This yields a simulated next state, $s_{t+1}$. The planner then recalculates the similarity weights for this new state, $w_i(s_{t+1})$, and evaluates the state-uncertainty $H(s_{t+1})$ and goal-likelihood $L(s_{t+1})$. A state is considered terminal if the uncertainty $H(s)$ falls below the threshold $\epsilon$, and the planner receives a negative reward if the feasibility constraint $L(s) < \tau$ is violated at any step. The immediate step reward, $R(s_t, A_t)$, is also recorded. This process allows the MCTS to build a search tree that accurately reflects the trade-off between reward (resource efficiency) and the expected information gain towards desired states, all guided by the historical data.

To mitigate stochasticity, we run the planning process multiple times to form an ensemble. The final recommendation (MLASP) is constructed by majority vote over the actions recommended by the ensemble policies at each stage. This ensures the plan is robust and not an artifact of a single simulation run.

## 5    EXPERIMENTS

We validate the performance of IBMDP through a two-part evaluation. First, we apply it to a real-world sequential assay planning task in central nervous system (CNS) drug discovery to demonstrate its practical utility and potential for resource savings. For reproducibility, we performed the same experiment on a public dataset on selecting *in vivo* pharmacokinetics assays between rat and dog to determine *in vivo* clearance in human (Appendix E). Second, we set up a synthetic environment with a known optimal policy to rigorously assess the quality of its decision-making process.

### 5.1    BRAIN PENETRATION ASSAYS: A REAL-WORLD CASE STUDY

**Problem Setting and Data.**    We evaluate IBMDP on a sequential assay-planning task for central nervous system (CNS) drug discovery, where the objective is to efficiently determine a compound's brain penetration potential. This property is critically dependent on the compound's ability to cross the blood-brain barrier (BBB). The decision involves selecting from cheap, fast, but less informative *in vitro* transporter assays (P-glycoprotein, PgP; Breast Cancer Resistance Protein, BCRP) and a definitive but slow and expensive *in vivo* assay that measures the unbound brain-to-plasma partition coefficient ($k_{\text{puu}}$).

Our historical dataset, $\mathcal{D}$, comprises $N = 220$ compounds with complete measurements for all relevant assays (100 nM PgP, 1 µM PgP, 100 nM BCRP) and the target property, $k_{\text{puu}}$. All compounds also have associated QSAR predictions, which provide initial estimates for the assay outcomes (e.g., for PgP and BCRP activity) and other relevant properties such as Mean Residence Time (MRT). The operational costs are defined as \$400 and a 7-day turnaround for each *in vitro* assay, and \$4,000 and a 21-day turnaround for the *in vivo* $k_{\text{puu}}$ assay. Actions are constrained to a maximum of 3 parallel assays per step. A compound is considered to have high potential if its $k_{\text{puu}} > 0.5$.

**Experimental Setup.**    The planning objective is to balance the reduction of state uncertainty $H(s)$ on the target $k_{\text{puu}}$, the increase in goal likelihood $L(s)$ that $k_{\text{puu}}$ is in the desirable range, and the maximization of reward (efficient use of resources). An experimental sequence terminates when the planner reaches a state of sufficient confidence, defined by the joint criteria $H(s_T) \leq \epsilon$ and $L(s_t) \geq \tau$ for all intermediate steps. Outcomes are compared against a conventional, rule-based decision strategy.

**Rule-Based Baseline.**    In practice, decisions often follow simple heuristics based on QSAR predictions. For this task, the baseline heuristic is:

- A compound is deemed **promising** (likely $0.5 \leq k_{\text{puu}} \leq 1$) if $\text{QSAR}_{\text{1uM\_PgP}} < 2$ AND $\text{QSAR}_{\text{100nM\_BCRP}} < 2$.

- A compound is deemed **non-promising** (likely $k_{\text{puu}} \leq 0.5$) if $\text{QSAR}_{\text{1uM\_PgP}} > 4$ OR $\text{QSAR}_{\text{100nM\_BCRP}} > 4$.

We evaluated IBMDP across four representative scenarios from three categories designed to test its performance against this heuristic: (i) **Baseline confirmation** (clear QSAR signals, Scenario 3), (ii) **Heuristic challenge** (conflicting or borderline QSAR signals, Scenarios 2 and 4), and (iii) **Opportunity discovery** (QSARs suggest a non-promising compound that is, in fact, good, Scenario 1).

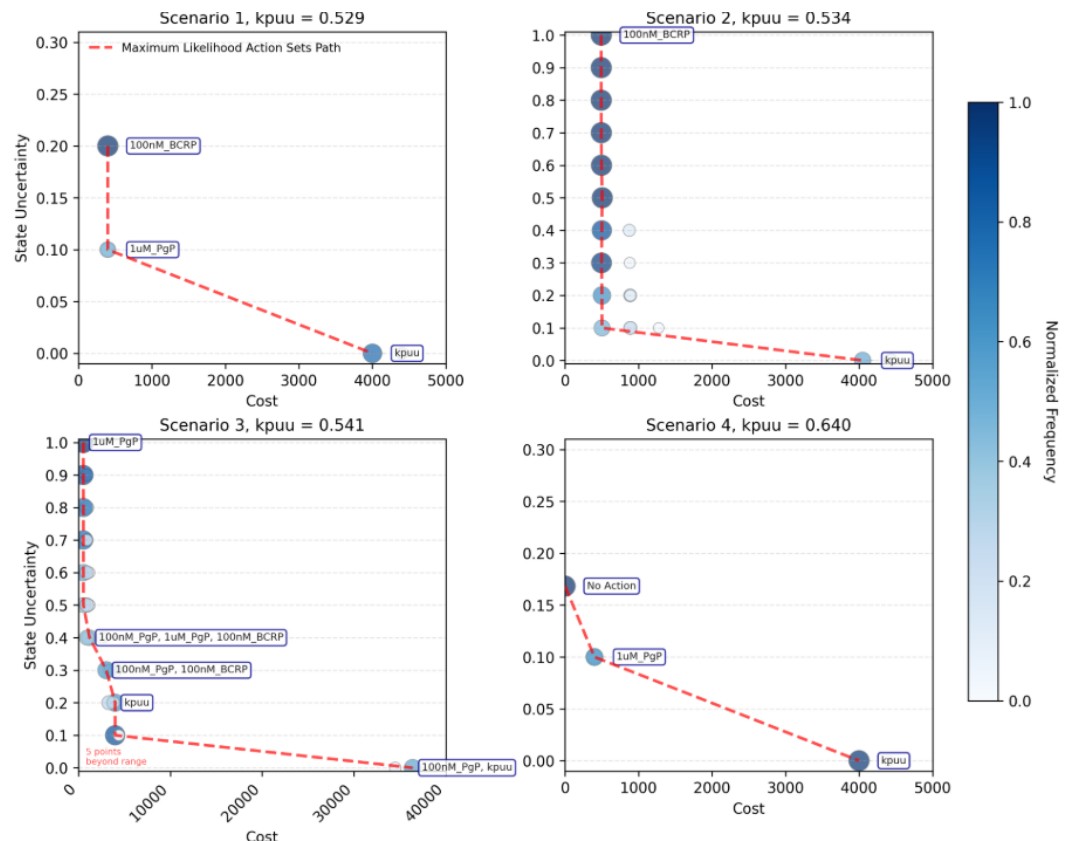

Figure 2: Monetary-prioritized results from IBMDP for four representative compounds. Each plot shows the Pareto front of achievable resource consumption versus terminal state uncertainty, with the Maximum-Likelihood Action-Sets Path (MLASP) highlighted. This illustrates how IBMDP provides a trade-off curve, allowing decision-makers to select a plan based on their risk and budget tolerance.

**Results.** As shown in Table 1, IBMDP consistently identifies more resource-efficient experimental plans than the traditional approach, which often defaults to running a full panel of assays consuming $5,200. The table rows are ordered to correspond directly to the scenarios shown in Figure 2. In the opportunity discovery scenario (row 1/Scenario 1), the heuristic would have incorrectly discarded a valuable compound, whereas IBMDP recommends an efficient $800 plan to reveal its true potential. For the next compound (row 2/Scenario 2), IBMDP finds a minimal $400 plan to resolve uncertainty. In the baseline confirmation scenario (row 3/Scenario 3), IBMDP recommends just $400-$800 to confirm the promising profile. Finally, for the challenging case with conflicting QSARs (row 4/Scenario 4), IBMDP efficiently resolves uncertainty for $400-$800. Across these representative cases, IBMDP achieves the same or higher level of decision confidence with up to 92% reduction in resource consumption.

## 5.2 SIMULATION WITH SYNTHETIC DATA

**Benchmark Setup.** To rigorously assess the policy quality of IBMDP in a controlled setting, we benchmarked it using a synthetic dataset where a theoretically optimal policy is computable (full details in Appendix D). We established this optimal policy using Value Iteration with the true, analytic uncertainty dynamics (VI-Theo). We then compared IBMDP against both this VI-Theo baseline and a deterministic variant using the same similarity-based estimation as IBMDP, but planned with Value Iteration (VI-Sim).

**Results.** The results, summarized in Table 2, demonstrate the effectiveness of IBMDP's stochastic, ensemble-based planning. Over 100 independent trials, IBMDP's primary recommendation (Top 1) aligned with the optimal VI-Theo policy in 47% of cases. In contrast, the deterministic VI-Sim

Table 1: Resource expense comparison between the traditional heuristic approach and IBMDP for representative compounds. Rows are ordered to match scenarios 1-4 in Figure 2. The traditional approach expense of $5200 reflects running the full assay panel ($4000 for $k_{\text{puu}}$ plus $3 \times \$400$ for *in vitro* assays), which IBMDP consistently avoids.

| QSAR Predictor | | | Assays | | | | Expense ($\times$\$100) | |
|---|---|---|---|---|---|---|---|---|
| 1uM PgP | 100nM BCRP | mrt | kpuu | 100nM PgP | 1uM PgP | 100nM BCRP | Trad. | IBMDP |
| 5.0 | 9.6 | 1.0 | 0.53 | 15.9 | 12.9 | 8.2 | 52 | 8 |
| 0.9 | 8.5 | 2.6 | 0.53 | 2.2 | 1.1 | 14.2 | 52 | 4 |
| 1.7 | 1.3 | 1.8 | 0.54 | 1.1 | 0.8 | 1.3 | 52 | 4 - 8 |
| 21.4 | 0.7 | 1.2 | 0.64 | 17.4 | 19.7 | 0.8 | 52 | 4 - 8 |

approach achieved only 36% alignment. The advantage of the ensemble approach is further highlighted by the fact that the optimal action was contained within IBMDP's top two recommendations 66% of the time, providing robust and effective coverage of the high-value policy space.

Table 2: Policy Alignment with Theoretical Baseline

| Method | Matches | Match Rate (%) |
|---|---|---|
| IBMDP Top 1 | 47 | 47.0 |
| IBMDP Top 2 | 66 | 66.0 |
| VI Similarity | 36 | 36.0 |

This superior performance stems from a fundamental difference in policy generation. While VI-based methods converge to a single, deterministic policy, IBMDP's ensemble of MCTS agents explores the policy space more broadly. This allows it to identify multiple, often near-equivalent, high-value actions, which is particularly advantageous in assay selection where different feature combinations can yield similar information gains. The results confirm that our ensemble-based planner provides more robust and reliable recommendations than a deterministic alternative by effectively navigating the uncertainty inherent in the policy space itself.

## 6 Conclusions

To achieve case-guided planning, we presented IBMDP, a reinforcement learning framework that turns historical cases into a generative model for sequential assay selection. By weighting historical cases based on similarity, the algorithm enables robust, multi-step planning with Monte Carlo Tree Search without requiring an explicit transition function. The application to a real-world drug discovery problem demonstrated it uncovers ground truth of a compound with fewer, cheaper assays. This work establishes a powerful methodology for leveraging past experience to guide future experiments, with broad applicability in fields beyond drug discovery where historical data is abundant but mechanistic models are scarce.

## 7 Related Work

**MDPs and Model-Based RL.** MDPs formalize sequential decision-making (Puterman, 2014); model-based RL learns dynamics for planning (Sutton & Barto, 2018; Kaiser et al., 2019; Moerland et al., 2023). Kernel-based RL leverages similarity primarily for value approximation or smoothing learned transitions (Ormoneit & Sen, 2002; Kveton & Theocharous, 2012; Xu et al., 2007). IBMDP uses similarity to build a *generative*, nonparametric transition *without* $(s, a, s')$ tuples—sampling assay outcomes from historical records rather than learning explicit kernels over next-states.

**Bayesian RL and Bayesian Optimization.** BRL maintains posteriors over model parameters or values and samples explicit MDPs (e.g., PSRL) (Ghavamzadeh et al., 2015; Osband et al., 2013; Agrawal & Jia, 2017). BO targets one-shot improvement of objective functions (Griffiths & Hernández-Lobato, 2020; Gómez-Bombarelli et al., 2018). IBMDP avoids explicit parameter posteriors and performs *Bayesian case-based generation* via similarity-weighted reweighting of records, enabling *multi-step* planning with reward optimization and feasibility constraints.

**Bayesian Experimental Design and Implicit Models.** Canonical single-step BO/BED methods are myopic and assume an explicit likelihood or simulator(Chaloner & Verdinelli, 1995; Rainforth et al., 2024); implicit-BED handles intractable likelihoods with info-theoretic surrogates or policy learning (Kleinegesse & Gutmann, 2020; 2021; Ivanova et al., 2021). IBMDP embeds an implicit model inside an *RL planner* (MCTS-DPW), balancing reward, time, and feasibility—not solely information gain.

**Constrained MDPs and POMDPs.** CMDPs typically constrain cumulative costs (Achiam et al., 2017); our constraints target *state* properties (terminal uncertainty, per-step likelihood) enforced during planning. The setting is akin to POMDPs (Kaelbling et al., 1998); our similarity-weighted posterior over records acts as an *implicit belief*. While multi-step RL/POMDP solvers require simulators or $(s, a, s')$ tuples, IBMDP uses a *similarity-weighted, implicit generative model* built directly from historical assay profiles, preserving cross-assay dependence without learning explicit dynamics. A direct benchmark is therefore not strictly comparable without substantial adaptation: (i) redefining utilities over *assays* rather than inputs, (ii) adding *resource-aware* batching and a principled *stopping* rule aligned with our constraints on $H(s)$ and $L(s)$, and (iii) supplying a *posterior predictive* consistent with the no-simulator setting (Appendix A).

**Ensembles in RL.** Ensembles improve robustness and uncertainty estimates (Dietterich, 2000; Zhou, 2012; Wiering & Van Hasselt, 2008; Osband et al., 2016; Lakshminarayanan et al., 2017). IBMDP use ensembling pragmatically to stabilize stochastic planning.

**Application Context.** Prior RL in biomedicine focuses on trials or molecule generation/synthesis (Bennett & Hauser, 2013; Eghbali-Zarch et al., 2019; Abbas et al., 2007; Fard et al., 2018; Wang et al., 2021; Bengio et al., 2021; You et al., 2018; Zhou et al., 2019; Segler et al., 2018). Assay selection in early discovery remains underexplored. IBMDP supplies a practical planner that converts historical assay records into a coherent, generative transition model with operational constraints, addressing the "no $(s, a, s')$" regime typical of discovery.

**On Fair Comparison with Related Methods.** While the above methods appear relevant, direct benchmarking would be fundamentally unfair—each operates under different mathematical assumptions and problem formulations. Model-based RL requires $(s, a, s')$ data or simulators; Bayesian RL samples from parameter posteriors; BO performs single-step optimization; POMDPs need explicit transition models. IBMDP uniquely addresses the setting where only static historical outcomes exist, making these comparisons "apples to oranges." See Appendix C for detailed analysis.

## 8 LIMITATIONS

**Historical data coverage.** Effectiveness hinges on the quality/representativeness of $\mathcal{D}$; gaps or bias can yield suboptimal choices. Unlike model-free RL with exploration, similarity-based sampling cannot discover strategies absent from $\mathcal{D}$—though in discovery, stable physico-chemical regularities partly mitigate this risk.

**Similarity metric assumptions.** The exponential kernel over (normalized) Euclidean distances assumes these distances reflect assay behavior. Nonlinear/threshold biology may violate this; domain-tailored metrics may be required to capture structure–activity relations.

**Scalability.** The worst-case total complexity is $O(N_e \cdot n_{\text{itr}} \cdot \min(b^H, n_{\text{itr}}) \cdot |\mathcal{D}| \cdot d)$, where $b$ is the effective branching factor and $H$ is the maximum tree depth. The per-iteration cost is dominated by the similarity weight calculation ($O(|\mathcal{D}| \cdot d)$). Large datasets $|\mathcal{D}|$ or high feature dimensions $d$ can strain memory and compute, potentially requiring distributed infrastructure or data subsampling for enterprise-scale use.

**Hyperparameter sensitivity.** Performance depends on tuning $\{\lambda_w, \lambda_k, N_e, c, \epsilon, \tau\}$; robustness across programs may require expert priors or nontrivial validation budgets.

**Ethics Statement**

In accordance with the ICLR Code of Ethics, this work is intended to contribute positively to society by addressing a key challenge in pharmaceutical research, the resource waste due to inefficient decisions and the use of preclinical animals in drug discovery.

The primary goal of the proposed framework, the Implicit Bayesian Markov Decision Process (IB-MDP), is to enhance human well-being by making the drug discovery process more efficient. By optimizing the sequence of experimental assays, this research aims to reduce the significant monetary and time costs associated with developing new medicines. An ethical benefit of this approach is the potential to minimize harm by reducing the number of costly and lengthy *in vivo* animal assays, prioritizing such scarce resource for only the most promising compounds.

We are committed to upholding high standards of scientific excellence and transparency. The IB-MDP framework was rigorously evaluated on both a real-world central nervous system (CNS) drug discovery task and a synthetic environment where the optimal policy was computable, ensuring a thorough assessment of its performance. We have been transparent about the method's limitations, particularly its dependence on the quality and representativeness of the historical data used for planning. The main ethical consideration is that a biased or incomplete historical dataset could lead to suboptimal decisions, potentially resulting in missed opportunities or wasted resources.

The research utilizes preclinical data on chemical compounds and their assay outcomes. It does not involve data from human subjects, thereby minimizing concerns related to personal privacy. We believe this work represents a responsible application of machine learning to a critical scientific domain.

**Reproducibility Statement**

To ensure the reproducibility of our findings, we have provided detailed descriptions of our methodology and experimental setup. The IBMDP framework is outlined in Section 4, with a concrete implementation provided in Algorithm 1. The theoretical underpinnings of our similarity-based model, including its formal correspondence to a POMDP, are detailed in Appendix A. For our theoretical claims, a complete derivation and consistency proof for the similarity-based estimator in the synthetic setting is available in Appendix D.

All experimental setups are described in Section 5. Full details on hyperparameter selection can be found in Appendix B. The process for generating the synthetic dataset is specified in Appendix D.1, and the public dataset used for the clearance optimization benchmark is cited and described in Appendix E. The source code, data, and scripts to reproduce results have been uploaded to the Supplementary Material, which will be visible to reviewers and the public throughout and after the review period.

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

# Appendix

## A  THEORETICAL FRAMEWORK: IB-MDP AS A POMDP

This appendix provides a formal conceptual grounding for the IB-MDP framework. We demonstrate that our similarity-weighted, case-based approach is not an ad-hoc heuristic, but rather a computationally tractable implementation of Bayesian belief updating within a Partially Observable Markov Decision Process (POMDP) tailored for information-gathering problems.

### A.1  POMDP PRELIMINARIES

A POMDP is formally defined by the tuple $(\mathcal{S}, \mathcal{A}, \Omega, P, O, R, \gamma)$, where $\mathcal{S}$ is a set of hidden states, $\mathcal{A}$ is the set of actions, and $\Omega$ is the set of observations. Since the agent cannot observe the true state $s \in \mathcal{S}$, it maintains a belief state, $b_t(s)$, which is a probability distribution over $\mathcal{S}$. After taking action $A_t$ and receiving observation $\omega_t$, the belief is updated via the Bayes filter:

$$b_{t+1}(s') \;\propto\; O(\omega_t \mid s', A_t) \sum_{s \in \mathcal{S}} P(s' \mid s, A_t)\, b_t(s). \tag{8}$$

**The Information-Gathering Case.**  The sequential assay planning task is an instance of an *information-gathering* problem. The underlying intrinsic properties of the candidate compound $x_\star$ are fixed; performing an assay reveals information about these properties but does not change them. This corresponds to a static latent state, where the transition probability is an identity function: $P(s' \mid s, A_t) = \mathbf{1}[s' = s]$. In this common special case, the belief update from Equation equation 8 simplifies to the multiplicative Bayes' rule:

$$b_{t+1}(s) \;\propto\; O(\omega_t \mid s, A_t)\, b_t(s). \tag{9}$$

### A.2  THE IB-MDP LATENT INDEX MODEL AND ITS POMDP CORRESPONDENCE

To map our framework to a POMDP, we introduce a discrete latent variable $Z \in \{1, \ldots, N\}$, where each value $i$ corresponds to one of the historical records $D_i \in \mathcal{D}$. We treat $Z$ as the hidden state, representing the "true prototype" of our candidate compound $x_\star$ from among the known historical cases. The core idea is that by maintaining a belief over $Z$, we are implicitly maintaining a belief about the complete, unobserved profile of $x_\star$. The explicit correspondence is detailed in Table 3.

### A.3  EQUIVALENCE OF THE SIMILARITY UPDATE AND BAYESIAN FILTERING

With the mapping established, we now demonstrate that the similarity weight update mechanism in IB-MDP is a direct implementation of the Bayesian belief update from Equation equation 9.

Let the prior belief over the latent index before step $t$ be the weights $w_i(s_t) \equiv P(Z = i \mid s_t)$. Executing the assay batch $A_t$ yields the observation $\omega_t \equiv \{y_{\star,j}\}_{j \in A_t}$. By substituting the IB-MDP analogs into Equation equation 9, we derive the IB-MDP belief update rule for the weights:

$$w_i(s_{t+1}) = \frac{p(\omega_t \mid Z = i, A_t)\, w_i(s_t)}{\sum_{\ell=1}^{N} p(\omega_t \mid Z = \ell, A_t)\, w_\ell(s_t)}. \tag{10}$$

This confirms that the evolution of weights in IB-MDP is a principled Bayesian recursion.

**Connecting the Likelihood to the Similarity Kernel.**  The final step is to show that our specific implementation of similarity weights corresponds to a valid probabilistic likelihood model. If we model the likelihood of observing an assay outcome $y_a$ for the candidate with a Gaussian kernel centered on the historical value $y_{i,a}$:

$$p(y_a \mid Z = i, a) \;\propto\; \exp\!\left( -\frac{\lambda_a}{2} \frac{(y_a - y_{i,a})^2}{\sigma_a^2} \right),$$

and assume conditional independence of assays in a batch given the prototype $Z$ (a modeling assumption that enables tractable inference; while biochemical assays may exhibit correlations even given compound properties, our empirical results demonstrate robustness to violations of this assumption through the ensemble averaging mechanism), the joint likelihood for the observation $\omega_t$ is the product of individual likelihoods. Applying the Bayesian update recursively from a uniform

prior over all observed assays $\{y_a\}_{a \in M_t}$ yields a posterior over $Z$ that has the exact form of our similarity weights:

$$w_i(s_t) \propto \prod_{a \in M_t} p(y_a \mid Z = i, a) = \exp\left(-\frac{1}{2} \sum_{a \in M_t} \lambda_a \frac{(y_a - y_{i,a})^2}{\sigma_a^2}\right) \equiv \exp\left(-\lambda_w \, d(s_t, D_i)\right),$$

where we can identify the global temperature parameter $\lambda_w = \beta/2$ where $\beta$ is the inverse temperature of the tempered posterior. With $\beta = 1$ (standard posterior), we have $\lambda_w = 1/2$. Therefore, our similarity function is not an arbitrary heuristic but corresponds to a tempered Bayesian posterior over the latent historical prototypes.

### A.4 The Posterior Predictive Transition Model

The belief state (the weight vector $w(s_t)$) is used for planning. The transition model used to simulate future trajectories within the MCTS planner is derived by marginalizing over the uncertainty in the latent variable $Z$. The probability of transitioning to a next state $s_{t+1}$ is the posterior predictive distribution over outcomes, conditioned on the current belief:

$$P(s_{t+1} \mid s_t, A_t) = \sum_{i=1}^{N} P(s_{t+1} \mid Z = i, s_t, A_t) P(Z = i \mid s_t) \tag{11}$$

$$= \sum_{i=1}^{N} w_i(s_t) \, \delta_{s_t \oplus \{(a_j, y_{i,j})\}_{j \in A_t}}(s_{t+1}), \tag{12}$$

where $\delta_x(y)$ denotes the Dirac delta measure that equals 1 if $y = x$ and 0 otherwise. This confirms that our sampling mechanism—drawing a historical case $D_i$ according to the weights $w_i(s_t)$ and using its outcomes—is a principled way to sample from the posterior predictive distribution, allowing the planner to explore plausible future scenarios consistent with all evidence gathered so far.

### A.5 Implications and Summary

Framing IB-MDP within the POMDP context provides strong conceptual grounding and yields several key insights, summarized in Table 3.

- **Justification for Dynamics**: The changing similarity weights observed *within* an MCTS simulation are not arbitrary non-stationarity. They represent the agent's evolving belief state. As information is gathered (the state $s_t$ is augmented), the model used for subsequent predictions naturally and correctly changes, reflecting a refined belief.

- **Suitability of MCTS**: MCTS is well-suited for this task because it is a simulation-based planner designed to handle complex state spaces. It effectively explores the consequences of actions on the future belief state and its associated rewards without needing an explicit representation of the belief space itself.

- **Principled Approximation**: IB-MDP provides a practical, data-driven approximation to solving a formal POMDP. Its effectiveness relies on two key assumptions: the quality and coverage of the historical data $\mathcal{D}$, and the appropriateness of the chosen kernel (e.g., Gaussian) for the observation likelihood model. While the Gaussian kernel provides computational tractability and aligns with common assumptions about measurement noise in biochemical assays, we acknowledge that alternative kernels (e.g., Laplacian, Student-t) may better capture heavy-tailed distributions or outliers. The robustness of our approach to kernel choice remains an important area for future empirical validation.

- **Computational Efficiency**: By representing the belief state implicitly through weights over the case-base $\mathcal{D}$, IB-MDP avoids the intractable calculations of maintaining and updating an explicit probability distribution over a potentially vast hidden state space.

In conclusion, interpreting IB-MDP as an approximate POMDP framework clarifies that the recalculation of similarity weights is a direct implementation of Bayesian belief updating. This justifies our methodology and the use of MCTS for principled planning under uncertainty when only historical data is available.

Table 3: Summary of the Conceptual Mapping between POMDP and IB-MDP.

| POMDP Component | IB-MDP Conceptual Equivalent | Notes |
|---|---|---|
| Hidden State ($s \in \mathcal{S}$) | Latent Index $Z = i$ over historical cases $D_i \in \mathcal{D}$ | The "true" but unknown profile of the candidate. |
| Belief State ($b_t(s)$) | Similarity weights $w_i(s_t) \equiv P(Z = i \mid s_t)$ | A probability distribution over possible prototypes. |
| Action ($A_t \in \mathcal{A}_t$) | Batch of assays to perform, $A_t \subseteq U_t$ | Direct equivalence. |
| Observation ($\omega_t$) | Set of assay outcomes $\{y_{\star,j}\}_{j \in A_t}$ | The new evidence gathered. |
| Observation Model ($O(\omega_t\|s', A_t)$) | Likelihood $p(\omega_t \mid Z = i, A_t)$ | Implemented via a similarity kernel. |
| Belief Update | Recalculation of weights $w_i(s_{t+1})$ | A direct, principled Bayesian update. |

## A.6 CONVERGENCE GUARANTEES AND THEORETICAL CONSIDERATIONS

The convergence properties of IBMDP differ fundamentally from traditional Bayesian reinforcement learning due to its unique reliance on historical data rather than environment interaction. This subsection examines what convergence guarantees can and cannot be provided.

### A.6.1 TRADITIONAL BAYESIAN RL GUARANTEES

Methods such as Posterior Sampling for Reinforcement Learning (PSRL) provide formal regret bounds of $\tilde{O}(\sqrt{SAT})$ for finite MDPs, where $S$ denotes states, $A$ denotes actions, and $T$ denotes the horizon. These approaches guarantee PAC-style convergence to $\epsilon$-optimal policies with high probability, leveraging the principle that posteriors concentrate on the true MDP as data accumulates.

### A.6.2 IBMDP CONVERGENCE PROPERTIES

IBMDP's convergence behavior is more nuanced due to its implicit model construction:

**Achievable Guarantees.** Standard MCTS with UCB1 provides asymptotic convergence to optimal policies as iterations approach infinity, assuming a fixed MDP model. For IBMDP specifically:

- MCTS-DPW convergence: The Double Progressive Widening variant used in IBMDP maintains convergence properties for large combinatorial action spaces

- Linear case consistency: For synthetic data with linear relationships and independent features, we prove (Section D) that the similarity-based variance estimator converges in probability to the true conditional variance as $N \to \infty$

- Empirical robustness: The ensemble approach with majority voting provides stable recommendations, achieving 47% optimal policy alignment compared to 36% for deterministic methods

**Fundamental Limitations.** Unlike traditional Bayesian RL, IBMDP cannot provide:

- Formal regret bounds: The implicit model introduces approximation error bounded by historical data coverage rather than converging to true dynamics

- PAC guarantees: Cannot ensure $\epsilon$-optimality with high probability due to dependence on data representativeness

- True dynamics recovery: The similarity-based model approximates but does not learn the true transition function $P(s'|s, a)$

The approximation quality depends on three key factors: (i) the coverage and representativeness of historical data $\mathcal{D}$, (ii) the appropriateness of the similarity metric for the domain, and (iii) the size of the historical dataset $|\mathcal{D}|$. While formal convergence rates cannot be established without access to the true dynamics, empirical validation demonstrates robust performance when these factors are satisfied.

**Convergence Trade-offs.** IBMDP trades formal convergence guarantees for practical applicability. Under assumptions of sufficient data coverage, appropriate similarity metrics, and regularity conditions, as MCTS iterations $n_{\text{itr}} \to \infty$ and ensemble size $N_e \to \infty$:

$$||\pi_{\text{IBMDP}} - \pi^*_{\text{empirical}}||_\infty \xrightarrow{P} 0 \tag{13}$$

where $\pi^*_{\text{empirical}}$ is the optimal policy for the empirical MDP induced by $\mathcal{D}$. However, the gap between this empirical optimal and the true optimal policy depends on data quality and coverage—a fundamental limitation when operating without simulators.

This trade-off is not a weakness but a necessary adaptation: IBMDP provides a principled solution where traditional methods with stronger guarantees cannot operate at all due to the absence of environment interaction capabilities.

# B  IMPLEMENTATION AND ALGORITHMIC DETAILS

## B.1  HYPERPARAMETER SELECTION METHODOLOGY

The performance of the IB-MDP framework depends on a set of key hyperparameters that govern the similarity model, the MCTS planner, and the problem's objective constraints. The values used in our experiments were determined through a combination of established literature guidelines, empirical testing on our specific dataset, and domain-specific considerations to balance decision quality with computational feasibility.

**Similarity Model Parameters.** These parameters define the core of the implicit, generative transition model.

- **Similarity Bandwidth ($\lambda_w$):** This parameter controls the "smoothness" of the similarity function. From the theoretical derivation in Section A, $\lambda_w = \beta/2$ where $\beta$ is the inverse temperature. For the standard posterior ($\beta = 1$), we have $\lambda_w = 0.5$. However, in practice, we tested values in the range $[0.5, 2.0]$ and found that $\lambda_w = 1.0$ provided better empirical performance for our dataset ($|\mathcal{D}| = 220, d = 6$). This corresponds to a tempered posterior with $\beta = 2.0$, which was large enough to ensure locality (giving higher weight to truly similar compounds) but small enough to draw support from a sufficient number of historical examples to make robust predictions.

- **Feature Weights ($\lambda_k = 1.0$ for all $k$):** These weights allow for emphasizing more or less informative features in the distance calculation. As we lacked detailed prior information on the relative reliability of the QSAR predictions and assay measurements, we set all weights to be equal to avoid introducing subjective bias. This treats all known features as equally important for determining similarity.

**Ensemble and Planner Parameters.** These parameters control the MCTS search algorithm and the robustness of its final policy.

- **Ensemble Size ($N_e = 50$):** To mitigate variance from the stochastic nature of both the transition model and the MCTS planner, we use an ensemble of independent runs. We tested sizes from 20 to 100 and found that $N_e = 50$ provided a stable policy recommendation (i.e., a consistent MLASP) without incurring excessive computational cost. Figure 3 illustrates how the ensemble's majority vote leads to a robust action choice.

- **MCTS Iterations ($n_{\text{itr}} = 20,000$):** This determines the search budget for each MCTS run. Our analysis showed that policy recommendations stabilized around 20,000 iterations for our problem's complexity, with diminishing returns for higher values.

- **Exploration Constant ($c = 5.0$):** Following standard practice for MCTS, this value balances exploration of new actions with exploitation of known high-value actions within the search tree. A value of 5.0 provided effective exploration in our experiments.

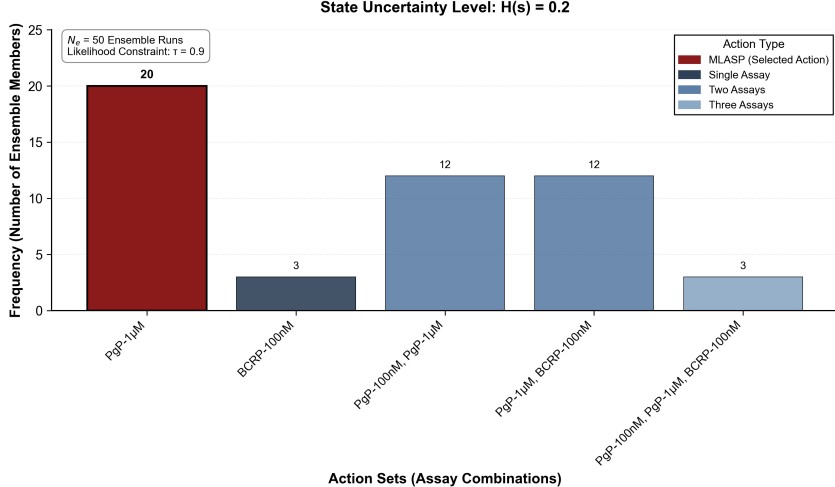

Figure 3: Example histogram of actions proposed across an ensemble of $N_e = 50$ runs. For a given state with uncertainty $\mathcal{H}(s) = 0.2$ and a likelihood constraint of $\tau = 0.9$, the action with the highest frequency is selected for the MLASP. This demonstrates how the ensemble method produces robust and stable recommendations via majority voting.

**Problem Constraint Parameters.** These parameters define the termination conditions and feasibility constraints of the planning problem itself.

- **Terminal Uncertainty Threshold ($\epsilon = 0.10$):** We stop when $H(s_T)$ drops below $0.10$. In our runs the initial uncertainty is between $0.2$ and $0.6$, so this threshold guarantees at least a two- to six-fold reduction before declaring the policy sufficiently confident.

- **Goal-Likelihood Threshold ($\tau \in \{0.6, 0.9\}$):** The threshold on the goal likelihood, $L(s_t)$, enforces that the planner only pursues trajectories that remain sufficiently likely to succeed. We tested two values to explore the trade-off between cost and confidence. A lower value ($\tau = 0.6$) permits more exploratory, potentially cheaper plans, while a higher value ($\tau = 0.9$) enforces a more conservative and confident, but potentially more expensive, policy. Figure 4 explicitly illustrates how a higher $\tau$ leads to a different and more costly MLASP to satisfy the stricter confidence requirement.

## B.2    COMPUTATIONAL COMPLEXITY ANALYSIS

The computational complexity of the IB-MDP algorithm is a critical factor for its practical application. The worst-case total complexity is given by:

$$O(N_e \cdot n_{\text{itr}} \cdot \min(b^H, n_{\text{itr}}) \cdot |\mathcal{D}| \cdot d),$$

where $N_e$ is the ensemble size, $n_{\text{itr}}$ is the number of MCTS iterations, $b$ is the effective branching factor (average number of actions explored per node), $H$ is the maximum tree depth (bounded by the horizon $T$), $|\mathcal{D}|$ is the number of historical cases, and $d$ is the dimensionality of the feature space. The term $\min(b^H, n_{\text{itr}})$ represents the maximum number of nodes that can be expanded, bounded either by the tree structure or the iteration budget. In practice, with progressive widening and UCT selection, the effective number of expansions is often much smaller than this worst-case bound.

The dominant factor within a single MCTS simulation step is the calculation of the similarity weights, which requires computing the distance from the current state to every historical case in $\mathcal{D}$. This operation has a complexity of $O(|\mathcal{D}| \cdot d)$ and is performed at each node expansion in the search tree.

**Comparison to Alternatives.** This complexity, while significant, compares favorably to alternative approaches for principled planning under uncertainty. Exact POMDP solvers are computationally intractable for problems of this scale, as their complexity is exponential in the size of the belief space. Traditional value iteration would require discretizing the state space, which becomes in-

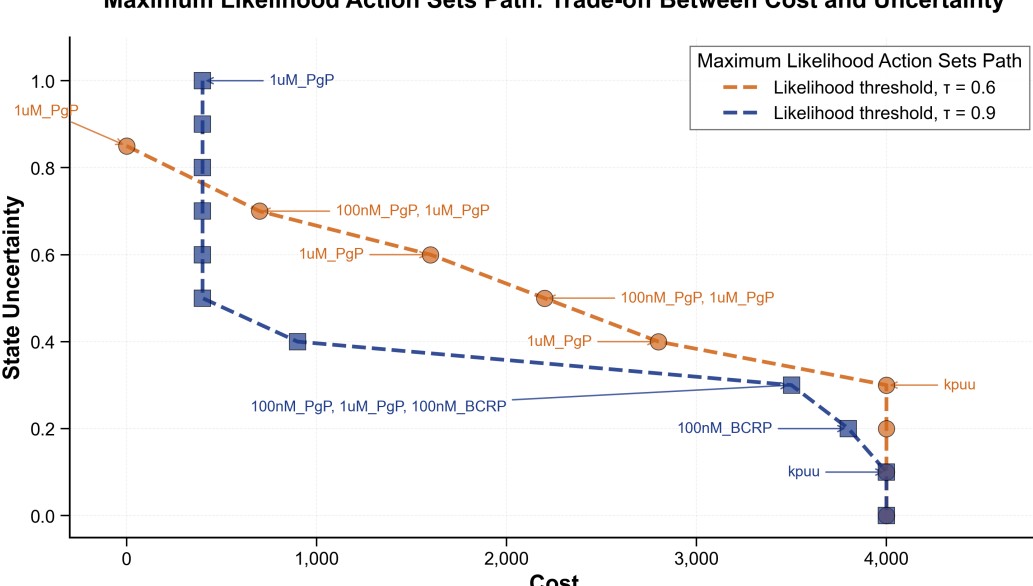

Figure 4: Comparison of MLASP paths for the same compound under two different goal-likelihood thresholds: $\tau = 0.6$ (blue) and $\tau = 0.9$ (red). The stricter constraint ($\tau = 0.9$) forces the planner to recommend a more expensive sequence of assays to achieve higher confidence, illustrating the direct trade-off between cost and decision confidence controlled by this parameter.

feasible with a growing number of continuous-valued assays. IB-MDP's sampling-based approach effectively navigates this high-dimensional space without requiring explicit enumeration.

**Practical Performance and Scalability.** In our experimental setup ($N_e = 50$, $n_{\mathrm{itr}} = 20{,}000$, $|\mathcal{D}| = 220$, $d = 6$), the total time to generate a policy for a single compound was approximately one hour on an Apple M1 Pro chip with 16GB of memory. The algorithm's complexity scales linearly with the size of the historical dataset ($|\mathcal{D}|$), the feature dimension ($d$), and the number of ensemble runs ($N_e$). This predictable scaling suggests that the method remains computationally feasible for the larger datasets and higher-dimensional problems typically encountered in real-world drug discovery campaigns, especially with access to parallel computing resources.

## C FRAMEWORK DIFFERENTIATION AND THE UNFAIRNESS OF DIRECT COMPARISON

### C.1 KEY DIFFERENTIATING FEATURES

Table 4 summarizes the fundamental distinctions between IBMDP and traditional reinforcement learning frameworks. These differences stem from IBMDP's unique design for sequential experimental planning in simulator-free, data-rich environments—a problem class that existing methods cannot address without fundamental restructuring.

### C.2 THE FUNDAMENTAL INNOVATION

IBMDP operates in an entirely different problem setting from traditional reinforcement learning. While conventional frameworks assume access to either environment simulators or transition data, IBMDP functions with only a static database of historical experimental outcomes. This constraint, common in drug discovery where mechanistic models are unavailable and experiments are irreversible, necessitates a fundamentally different approach.

Table 4: Fundamental distinctions between IBMDP and traditional RL frameworks

| Aspect | Traditional Frameworks | IBMDP |
|---|---|---|
| **Data Requirements** | $(s, a, s')$ tuples or simulator | Static historical outcomes only |
| **Transition Model** | Learned explicit $P(s'|s, a)$ | Implicit via similarity sampling |
| **Belief Representation** | Explicit probability distributions | Similarity weights $w_i(s_t)$ |
| **Planning Method** | Single policy or parametric | Ensemble MCTS with majority voting |
| **Constraints** | Cumulative: $\sum_t c_t \leq C$ | State-based: $H(s_T) \leq \epsilon, L(s_t) \geq \tau$ |
| **Action Space** | Parameter optimization | Combinatorial assay selection |
| **Action Effect** | Changes underlying state | Reveals fixed properties |
| **Correlation Handling** | Requires explicit modeling | Preserves empirically via sampling |
| **Objective** | Single reward maximization | Multi-objective optimization |

The core mechanism constructs an implicit generative model through similarity-weighted sampling:

$$w_i(s_t) = \exp\left(-\lambda_w \cdot d(s_t, D_i)\right) \tag{14}$$

$$P(s_{t+1}|s_t, A_t) = \sum_{i=1}^{N} \frac{w_i(s_t)}{Z} \cdot \mathbf{1}[s_{t+1} = s_t \oplus \{(a_j, y_{i,j})\}_{j \in A_t}] \tag{15}$$

This mechanism generates plausible transitions by sampling from historical cases most similar to the current experimental state, thereby preserving the natural correlations between assays observed in real compounds—correlations that would be difficult or impossible to model explicitly given current scientific understanding.

### C.3 COMPARISON WITH EXISTING FRAMEWORK CATEGORIES

#### C.3.1 DISTINCTION FROM MDPS AND MODEL-BASED RL

Model-based reinforcement learning fundamentally relies on learning transition dynamics from $(s, a, s')$ tuples, typically through parametric models that approximate $P(s'|s, a)$. Even kernel-based RL methods, which employ similarity metrics, use them primarily for value function approximation rather than transition generation.

IBMDP diverges by using similarity not as a smoothing mechanism but as the foundation for a complete generative process. Without access to any transition data, it samples entire assay outcome profiles from historical compounds, weighted by their relevance to the current state. This non-parametric approach sidesteps the need for explicit dynamics modeling while naturally preserving cross-assay dependencies present in the historical data.

#### C.3.2 DISTINCTION FROM BAYESIAN METHODS

Bayesian reinforcement learning and Bayesian optimization maintain explicit posterior distributions—over model parameters in BRL (exemplified by PSRL) or over objective functions in BO. These methods either sample complete MDPs from parameter posteriors or perform myopic single-step optimization.

IBMDP performs what we term Bayesian case-based generation: the similarity weights serve as an implicit posterior over historical compound prototypes, updated through reweighting as evidence accumulates. Unlike BO's single-step focus, IBMDP enables multi-horizon planning that simultaneously considers experimental costs, time constraints, and the probability of achieving desired outcomes—a multi-objective optimization fundamentally different from traditional Bayesian approaches.

#### C.3.3 DISTINCTION FROM EXPERIMENTAL DESIGN

Classical Bayesian experimental design assumes availability of a likelihood function or simulator, optimizing for immediate information gain. Even implicit-BED methods for intractable likelihoods rely on information-theoretic surrogates that assume some form of generative model.

IBMDP embeds its implicit model directly within a reinforcement learning planner (MCTS-DPW), optimizing entire experimental sequences rather than individual experiments. The framework's state-uncertainty functional $H(s_t)$ and goal-likelihood functional $L(s_t)$ provide interpretable,

domain-specific measures that directly relate to experimental objectives, unlike abstract information-theoretic quantities.

### C.3.4 DISTINCTION FROM POMDPs

Standard POMDP formulations maintain explicit belief distributions over hidden states, requiring specification of both transition models $P(s'|s, a)$ and observation models $O(o|s, a)$. The belief update follows the Bayes filter, necessitating these explicit models.

IBMDP's similarity-weighted posterior serves as an implicit belief representation, eliminating the need for high-dimensional belief state maintenance. The framework's constraints—terminal uncertainty $H(s_T) \leq \epsilon$ and per-step feasibility $L(s_t) \geq \tau$—target state properties rather than cumulative quantities, directly encoding experimental requirements for decision confidence and trajectory viability.

### C.4 WHY DIRECT BENCHMARKING IS FUNDAMENTALLY UNFAIR

The fundamental incompatibility between IBMDP and traditional frameworks makes direct benchmarking not merely challenging but inherently unfair—comparing methods designed for entirely different problem settings and data availability.

### C.4.1 INCOMPATIBLE PREREQUISITES

Traditional RL methods universally require either an environment simulator for generating transitions on demand or a collection of $(s, a, s')$ tuples for learning dynamics. IBMDP operates precisely where these prerequisites are absent: only static historical compound profiles exist, with no mechanism to query counterfactual outcomes. Creating a simulator would require mechanistic understanding of biochemical interactions that current science lacks, while collecting transition data through exhaustive experimentation defeats the very purpose of efficient planning.

### C.4.2 FUNDAMENTAL STRUCTURAL DIFFERENCES

The action spaces are categorically different. Traditional methods optimize over continuous or discrete parameter spaces where actions affect state transitions. IBMDP selects from combinatorial sets of experimental assays—$\mathcal{P}_{\leq m}(U_t) \cup \{\text{eox}\}$—where actions reveal information about unchanging molecular properties. This distinction between control and information gathering necessitates entirely different planning paradigms.

Furthermore, the constraint structures are incompatible. Traditional constrained MDPs limit cumulative costs across trajectories, while IBMDP enforces instantaneous feasibility constraints and terminal uncertainty bounds that directly encode experimental requirements.

### C.4.3 REQUIRED TRANSFORMATIONS

Adapting traditional methods to this setting would require:

1. Completely redefining the action space from parameter optimization to combinatorial assay selection with batching constraints

2. Implementing reward-aware stopping rules aligned with uncertainty and feasibility functionals rather than simple cumulative objectives

3. Creating posterior predictive distributions without access to simulators or transition data

4. Restructuring from single-objective to multi-objective optimization with state-based constraints

Such extensive modifications would fundamentally alter the nature of these methods, creating essentially new algorithms rather than variants of existing ones. Any resulting comparison would be between IBMDP and these newly created methods, not the original frameworks.

## C.5 THEORETICAL FOUNDATION

Despite operating in this unique setting, IBMDP maintains rigorous theoretical grounding. Section A establishes that the framework is mathematically equivalent to a POMDP where the hidden state represents a latent index over historical cases. The similarity weights constitute a valid Bayesian posterior, with weight updates implementing exact Bayesian belief updates. This equivalence:

$$w_i(s_{t+1}) = \frac{p(\omega_t|Z = i, A_t) \cdot w_i(s_t)}{\sum_{\ell=1}^{N} p(\omega_t|Z = \ell, A_t) \cdot w_\ell(s_t)} \tag{16}$$

provides principled justification for the empirical success observed in our experiments, where IBMDP achieved up to 92% reduction in resource consumption while maintaining decision quality.

## C.6 IMPLICATIONS

IBMDP addresses a problem class—sequential experimental planning without simulators—that existing reinforcement learning frameworks were not designed to handle. The inherent unfairness of direct benchmarking reflects not a limitation but the framework's fundamental novelty operating in a unique problem setting. By leveraging historical data through similarity-weighted sampling and ensemble planning, IBMDP provides the first practical solution for case-guided sequential decision-making in drug discovery and similar experimental sciences where traditional RL assumptions fail to hold.

# D BENCHMARK WITH SYNTHETIC DATA

**Overview and Motivation.** This appendix presents a rigorous benchmark study comparing IBMDP against theoretically optimal and deterministic baselines using synthetic data. The synthetic environment provides a unique advantage: we can compute the true optimal policy exactly, enabling principled validation of our similarity-based planning approach. This controlled setting allows us to isolate and evaluate the effectiveness of IBMDP's core innovations—the similarity-weighted belief mechanism and ensemble planning—against ground truth.

**Aim.** We provide a controlled benchmark to compare three planners for sequential assay selection: (i) a *theoretical* Value Iteration baseline with *exact* uncertainty dynamics (**VI-Theo**); (ii) a *deterministic* Value Iteration with *similarity-based* uncertainty (**VI-Sim**); and (iii) the *stochastic* IBMDP planner using *similarity-weighted posterior predictive* transitions inside an ensemble of MCTS-DPW trees (**IBMDP**). A synthetic data generator with known structure enables exact computation of the VI-Theo policy and a principled testbed for the two similarity-based planners.

**Notation consistency with the main text.** We maintain consistency with the notation from the main exposition (state/action tuples, similarity weights, $H(s)$, $L(s)$, etc.; see Equations equation 1–equation 3 and Table 8). Throughout this appendix, for notational convenience when the focus is on the set structure rather than individual measurements, we may write the state as $s = (x_\star, \mathcal{M})$ where $x_\star$ is the fixed candidate compound and $\mathcal{M} \subseteq \{1, \ldots, M\}$ represents the set of measured assays. This is equivalent to the main text notation $s_t = (x_\star, \{y_{\star,j}\}_{j \in M_t})$ where $\mathcal{M} = M_t$ indexes the measured assays and their values. When the candidate is fixed and clear from context, we may write the state simply as $\mathcal{M}$.

## D.1 SYNTHETIC DATA: GENERAL MODEL AND INSTANTIATION

**Purpose.** We construct a synthetic data environment where the true conditional variance can be computed analytically, providing ground truth for evaluating our similarity-based estimators. The linear structure with independent features represents a simplified but informative test case where theoretical optimality is tractable.

### D.1.1 GENERAL DATA-GENERATING PROCESS (GENERIC FORMULATION).

Fix integers $N$ (number of historical cases) and $M$ (number of assays/features). For each historical case $i \in \{1, \ldots, N\}$ we draw a feature vector

$$\mathbf{y}_i = (y_{i,1}, \ldots, y_{i,M})^\top \in \mathbb{R}^M.$$

For each assay $a \in \{1, \ldots, M\}$ specify distributional parameters $(\mu_a, \sigma_a, a_a, b_a)$ and draw independently

$$y_{i,a} \sim \mathcal{TN}(\mu_a, \sigma_a;\ a_a, b_a),$$

the (univariate) truncated normal on $[a_a, b_a]$ with location $\mu_a$ and scale $\sigma_a$.[1] Let $\beta = (\beta_1, \ldots, \beta_M)^\top \in \mathbb{R}^M$ and draw independent measurement noise

$$\epsilon_i \sim \mathcal{TN}(\mu_\epsilon, \sigma_\epsilon;\ a_\epsilon, b_\epsilon).$$

The scalar target is

$$g_i = \sum_{a=1}^{M} \beta_a\, y_{i,a} + \epsilon_i = \beta^\top \mathbf{y}_i + \epsilon_i.$$

The historical dataset is $\mathcal{D} = \{(x_i, \mathbf{y}_i)\}$; targets $G = \{g_i\}$ are stored separately (and *never* used in any distance/weight computation).

**Closed-form variance under independence.** Let $\Sigma = \mathrm{diag}(\sigma_1^2, \ldots, \sigma_M^2)$ denote the per-assay variance parameters (treated as the empirical variances of the generated $y_{i,a}$'s). Then

$$\mathrm{Var}(g) = \beta^\top \Sigma \beta + \sigma_\epsilon^2.$$

To derive the conditional variance, we partition assays into measured $\mathcal{M}$ and unmeasured $\mathcal{U}$ and write $\beta = (\beta_\mathcal{M}, \beta_\mathcal{U})$, $\Sigma = \mathrm{diag}(\Sigma_\mathcal{M}, \Sigma_\mathcal{U})$. Under independence, we have the following derivation:

$$\mathrm{Var}(g \mid \mathbf{y}_\mathcal{M}) = \mathrm{Var}\big(\beta_\mathcal{M}^\top \mathbf{y}_\mathcal{M} + \beta_\mathcal{U}^\top \mathbf{y}_\mathcal{U} + \epsilon \mid \mathbf{y}_\mathcal{M}\big) \tag{17}$$

$$= \mathrm{Var}\big(\beta_\mathcal{U}^\top \mathbf{y}_\mathcal{U} + \epsilon \mid \mathbf{y}_\mathcal{M}\big) \quad \text{(since } \beta_\mathcal{M}^\top \mathbf{y}_\mathcal{M} \text{ is fixed given } \mathbf{y}_\mathcal{M}) \tag{18}$$

$$= \mathrm{Var}\big(\beta_\mathcal{U}^\top \mathbf{y}_\mathcal{U}\big) + \mathrm{Var}(\epsilon) \quad \text{(by independence of } \mathbf{y}_\mathcal{U}, \epsilon \text{ from } \mathbf{y}_\mathcal{M}) \tag{19}$$

$$= \beta_\mathcal{U}^\top \mathrm{Var}(\mathbf{y}_\mathcal{U}) \beta_\mathcal{U} + \sigma_\epsilon^2 \tag{20}$$

$$= \beta_\mathcal{U}^\top \Sigma_\mathcal{U} \beta_\mathcal{U} + \sigma_\epsilon^2. \tag{21}$$

This identity is central to the VI-Theo derivation below.

D.1.2 Instantiation used in the benchmark.

We set $M = 6$ and $N = 200$. For $a = 1, \ldots, 6$,

$$\mu_a = 50 \cdot \frac{a}{6}, \qquad \sigma_a = 0.3\,\mu_a, \qquad (a_a, b_a) = (0,\, 2\mu_a),$$

$$\beta = (0.3,\, 0.25,\, 0.2,\, 0.15,\, 0.07,\, 0.03), \qquad \epsilon \sim \mathcal{TN}(0, 5;\, -10, 10).$$

All feature draws are independent across $a$ and $i$; noise is independent of features.

**Sampling recipe (for reproducibility).** For each trial: (1) fix $M, N, \{\mu_a, \sigma_a, a_a, b_a\}$ and $\beta$; (2) draw $\{\mathbf{y}_i\}_{i=1}^N$ componentwise; (3) draw $\{\epsilon_i\}_{i=1}^N$; (4) set $g_i = \beta^\top \mathbf{y}_i + \epsilon_i$; (5) store $\mathcal{D} = \{(x_i, \mathbf{y}_i)\}$ and $G = \{g_i\}$ with availability set $I_g = \{i : g_i \text{ used in evaluation}\}$.

D.2 Theoretical Baseline (VI-Theo): Full Derivation

**Overview.** VI-Theo represents the theoretically optimal policy under perfect information about the data-generating process. It uses the exact conditional variance formula derived above to compute optimal uncertainty reduction at each step. This baseline is only computable in synthetic settings where the true model parameters are known.

**State, action, and dynamics.** The state is the set $\mathcal{M} \subseteq \{1, \ldots, 6\}$ of measured assays (consistent with our notation convention). The action space is the power set of unmeasured assays:

$$A(\mathcal{M}) = \mathcal{P}\big(\{1, \ldots, 6\} \setminus \mathcal{M}\big).$$

Transitions are deterministic: executing $a \in A(\mathcal{M})$ yields $\mathcal{M} \leftarrow \mathcal{M} \cup a$.

---

[1] In our experiments we treat $(\mu_a, \sigma_a)$ as the empirical mean/scale of the generated samples; the truncation mildly perturbs the theoretical moments.

**Exact conditional variance and reduction.** Write $g = \beta_{\mathcal{M}}^\top \mathbf{y}_{\mathcal{M}} + \beta_{\mathcal{U}}^\top \mathbf{y}_{\mathcal{U}} + \epsilon$. Conditioning on the realized measurements $\mathbf{y}_{\mathcal{M}}$,

$$\mathrm{Var}(g \mid \mathbf{y}_{\mathcal{M}}) = \mathrm{Var}(\beta_{\mathcal{U}}^\top \mathbf{y}_{\mathcal{U}} + \epsilon) = \beta_{\mathcal{U}}^\top \Sigma_{\mathcal{U}} \beta_{\mathcal{U}} + \sigma_\epsilon^2,$$

since $\mathbf{y}_{\mathcal{U}}$ is independent of $\mathbf{y}_{\mathcal{M}}$ and $\epsilon$. Hence, the *exact* uncertainty reduction achieved by measuring a batch $a \subseteq \mathcal{U}$ is computed as follows:

$$\Delta\sigma_a^2 = \mathrm{Var}(g \mid \mathbf{y}_{\mathcal{M}}) - \mathrm{Var}(g \mid \mathbf{y}_{\mathcal{M} \cup a}) \tag{22}$$

$$= \left(\beta_{\mathcal{U}}^\top \Sigma_{\mathcal{U}} \beta_{\mathcal{U}} + \sigma_\epsilon^2\right) - \left(\beta_{\mathcal{U}\backslash a}^\top \Sigma_{\mathcal{U}\backslash a} \beta_{\mathcal{U}\backslash a} + \sigma_\epsilon^2\right) \tag{23}$$

$$= \beta_{\mathcal{U}}^\top \Sigma_{\mathcal{U}} \beta_{\mathcal{U}} - \beta_{\mathcal{U}\backslash a}^\top \Sigma_{\mathcal{U}\backslash a} \beta_{\mathcal{U}\backslash a} \tag{24}$$

$$= \sum_{k \in a} \beta_k^2 \sigma_k^2. \tag{25}$$

The last equality follows because $\Sigma$ is diagonal and the contribution of each measured assay $k$ is exactly $\beta_k^2 \sigma_k^2$.

**Costs and reward.** Let per-assay costs be

$$c_1 = 1.0, \quad c_2 = 1.2, \quad c_3 = 1.5, \quad c_4 = 1.8, \quad c_5 = 2.0, \quad c_6 = 2.2,$$

and (optionally) a terminal target-measurement cost $c_{\text{target}} = 10.0$. The batch cost is $c_a = \sum_{k \in a} c_k$ and the immediate reward is *uncertainty reduction per unit cost*:

$$R(\mathcal{M}, a) = \frac{\Delta\sigma_a^2}{c_a}.$$

**Bellman recursion.** With discount $\gamma = 0.95$,

$$V(\mathcal{M}) = \max_{a \in A(\mathcal{M})} \left\{ R(\mathcal{M}, a) + \gamma V(\mathcal{M} \cup a) \right\},$$

initialized at $V(\{1, \ldots, 6\}) = 0$ (no uncertainty left, no action left). We iterate to a tolerance of $10^{-6}$ or 1000 iterations to obtain the optimal policy $\pi_{\text{Theo}}(\mathcal{M})$.

**Remarks on optimality.** Because transitions are deterministic and rewards are additive with discount, the recursion gives the exact optimal policy under the synthetic uncertainty model. This policy serves as the *ground-truth baseline* against which we compare similarity-based planners.

### D.3 DETERMINISTIC SIMILARITY-BASED VI (VI-SIM): FULL DERIVATION

**Overview.** VI-Sim represents a deterministic planner that uses the same similarity-based variance estimation as IBMDP but applies Value Iteration instead of stochastic tree search. This baseline isolates the contribution of IBMDP's ensemble planning approach by using the same implicit model but with deterministic optimization.

**Similarity weights and distance.** At state $s = (x_\star, \mathcal{M})$ (maintaining our consistent notation; here we use $s$ to denote a generic state rather than $s_t$ for a specific time step), define:

$$w_i(s) = \frac{\exp\{-\lambda_w \, d(s, D_i)\}}{\sum_{j=1}^N \exp\{-\lambda_w \, d(s, D_j)\}}, \tag{26}$$

$$d(s, D_i) = \sum_{k \in \mathcal{K}(s)} \lambda_k \frac{\left(\phi_k(s) - \phi_k(D_i)\right)^2}{\sigma_k^2}. \tag{27}$$

Here $\mathcal{K}(s)$ are the known features (initial QSARs and any measured assays); $\phi_k(\cdot)$ extracts feature $k$; $\lambda_k$ are feature weights (default = 1), $\lambda_w > 0$ is the bandwidth, and $\sigma_k^2$ are the empirical variances over $\mathcal{D}$. *The targets $\{g_i\}$ are never used in $d(\cdot, \cdot)$.*

**Weighted $g$-mean and variance (renormalized over $I_g$).** Define the renormalized weights and weighted mean:

$$\tilde{w}_i(s) = \frac{w_i(s)}{\sum_{\ell \in I_g} w_\ell(s)} \quad \text{for } i \in I_g, \tag{28}$$

$$\bar{g}(s) = \sum_{i \in I_g} \tilde{w}_i(s)\, g_i. \tag{29}$$

The estimated conditional variance at state $s$ is:

$$\widehat{\sigma}^2_{\text{cond}}(\mathcal{M}) = \sum_{i \in I_g} \tilde{w}_i(s)\left(g_i - \bar{g}(s)\right)^2. \tag{30}$$

After executing batch $a$, we update the observed state to $s' = (x_\star, \mathcal{M} \cup a)$, recompute weights based on the expanded feature set, and obtain $\widehat{\sigma}^2_{\text{cond}}(\mathcal{M} \cup a)$.

**Estimated reduction and reward.** Using the similarity-weighted variance estimate, we define the variance reduction and reward:

$$\Delta\widehat{\sigma}^2_a = \widehat{\sigma}^2_{\text{cond}}(\mathcal{M}) - \widehat{\sigma}^2_{\text{cond}}(\mathcal{M} \cup a), \tag{31}$$

$$R(\mathcal{M}, a) = \frac{\Delta\widehat{\sigma}^2_a}{c_a}. \tag{32}$$

We then apply this $R$ to the same deterministic VI recursion as in Section D.2 to obtain the policy $\pi_{\text{Sim}}$.

### D.4 IBMDP: Implicit Posterior-Predictive Model and Planning Details

**Overview.** IBMDP extends the similarity-based approach with two key innovations: (1) a stochastic posterior-predictive model that samples from historical cases weighted by similarity, and (2) ensemble MCTS planning that explores multiple policy trajectories. This combination enables robust planning despite the implicit model's inherent uncertainty.

**Latent-index view and likelihood.** Introduce a discrete latent index $Z \in \{1, \ldots, N\}$ over historical cases $D_i$. Given $Z = i$ and selecting assay $a$, a Gaussian discrepancy model leads to:

$$p(y_a \mid Z = i, a) \propto \exp\left(-\frac{\lambda_a(y_a - y_{i,a})^2}{2\sigma_a^2}\right), \tag{33}$$

with per-assay weight $\lambda_a > 0$. For a batch $A_t$ and assuming conditional independence across assays given $Z$ (see discussion in Section A regarding this assumption):

$$p(\mathbf{y}_{A_t} \mid Z = i, A_t) \propto \prod_{a \in A_t} \exp\left(-\frac{\lambda_a(y_a - y_{i,a})^2}{2\sigma_a^2}\right). \tag{34}$$

**Weights as (tempered) posteriors and incremental recursion.** Let $O_t$ denote the set of observed assays up to time $t$. Define the variance-normalized distance

$$d(s_t, D_i) = \sum_{(a, y_a) \in O_t} \frac{\lambda_a}{\sigma_a^2}\left(y_a - y_{i,a}\right)^2.$$

With a uniform prior over $Z$ and temperature $\lambda_w$, the similarity weight equals a tempered posterior

$$w_i(s_t) = \frac{\exp\{-\lambda_w\, d(s_t, D_i)\}}{\sum_{j=1}^N \exp\{-\lambda_w\, d(s_t, D_j)\}}.$$

If we then measure assay $a_t$ and observe $y_{a_t}$, the distance updates additively:

$$d(s_{t+1}, D_i) = d(s_t, D_i) + \frac{\lambda_{a_t}}{\sigma_{a_t}^2}\left(y_{a_t} - y_{i,a_t}\right)^2. \tag{35}$$

This yields the multiplicative weight update:

$$w_i(s_{t+1}) \propto w_i(s_t) \cdot \exp\left\{-\lambda_w \frac{\lambda_{a_t}}{\sigma_{a_t}^2}\left(y_{a_t} - y_{i,a_t}\right)^2\right\} \tag{36}$$

$$\propto w_i(s_t) \cdot \left[p(y_{a_t} \mid Z = i, a_t)\right]^{2\lambda_w}, \quad \text{where } \lambda_w = \beta/2, \tag{37}$$

followed by normalization. Thus the reweighting is a (tempered) Bayesian belief update.

**Posterior predictive (implicit transition).** Marginalizing over $Z$ gives the posterior predictive over next information states:

$$P(s_{t+1} \mid s_t, A_t) = \sum_{i=1}^{N} w_i(s_t) \cdot \delta\Big(s_{t+1} = s_t \oplus \{(a, y_{i,a})\}_{a \in A_t}\Big), \tag{38}$$

which is implemented operationally by sampling a single historical case $i \sim \mathrm{Cat}(\{w_k(s_t)\})$ and *copying* the batch outcomes $\{y_{i,a}\}_{a \in A_t}$ into the candidate—thereby preserving cross-assay correlation within the sampled historical profile.

**Planning with MCTS-DPW and ensembling.** Within each MCTS rollout, we generate stochastic next states by the posterior predictive above, accrue step cost $R(s_t, A_t)$, and apply penalties when feasibility is violated (e.g., $L(s) < \tau$ at any step) until a terminal state (e.g., $H(s) \leq \epsilon$) or horizon $T$. To reduce variance from stochastic sampling and tree search, we run $N_e$ independent trees and aggregate recommendations by majority vote, reporting both the *Top-1* action and the *Top-2* action set at each decision step, forming an MLASP.

### D.5 ILLUSTRATIVE EXAMPLE: EVOLUTION OF SIMILARITY WEIGHTS

**Purpose.** This toy example demonstrates how similarity weights evolve as evidence accumulates, providing intuition for the adaptive nature of IBMDP's implicit dynamics.

**Setup.** Three historical records with one feature each at values $\{0, 1, 2\}$; let $\sigma^2 = 1$, $\lambda_w = \lambda_1 = 1$. The initial candidate state is $s^{(0)}$ with value 1.1.

**Step 0 (initial).** Raw weights:

$$w_1 = e^{-(1.1-0)^2} = e^{-1.21} = 0.297, \tag{39}$$

$$w_2 = e^{-(1.1-1)^2} = e^{-0.01} = 0.990, \tag{40}$$

$$w_3 = e^{-(1.1-2)^2} = e^{-0.81} = 0.445. \tag{41}$$

After normalization $Z = 0.297 + 0.990 + 0.445 = 1.732$, we obtain:

$$\tilde{w} = (0.171, 0.571, 0.257). \tag{42}$$

**Step 1 (after action moves state to 1.6).** Raw weights:

$$w_1 = e^{-(1.6-0)^2} = e^{-2.56} = 0.077, \tag{43}$$

$$w_2 = e^{-(1.6-1)^2} = e^{-0.36} = 0.697, \tag{44}$$

$$w_3 = e^{-(1.6-2)^2} = e^{-0.16} = 0.852. \tag{45}$$

Normalizing with $Z' = 1.626$ gives:

$$\tilde{w} = (0.047, 0.429, 0.524). \tag{46}$$

The posterior shifts toward the historical record at 2 as evidence moves rightward.

### D.6 THEORETICAL ANALYSIS: CONSISTENCY OF VI-SIM

**Theorem (Consistency of Similarity-Based Estimation).** Under the synthetic linear model with independent features, the similarity-based variance estimator $\widehat{\sigma}^2_{\mathrm{cond}}(\mathcal{M})$ used by VI-Sim converges in probability to the exact conditional variance $\sigma^2_{\mathrm{cond}}(\mathcal{M})$ used by VI-Theo, i.e., $\widehat{\sigma}^2_{\mathrm{cond}}(\mathcal{M}) \xrightarrow{P} \sigma^2_{\mathrm{cond}}(\mathcal{M})$ as $N \to \infty$.

**Proof.**

**Assumptions.** (A1) Features $\{y_{i,a}\}$ are independent across $a$ and i.i.d. across $i$, each with finite variance $\sigma^2_a$; (A2) noise $\epsilon_i$ is independent of features with finite variance $\sigma^2_\epsilon$; (A3) the weight function $w_i(s)$ depends only on *measured* assays $\mathcal{M}$ of each record and on the candidate's observed

values at those assays; (A4) the renormalized weights $\tilde{w}_i(s)$ over $I_g$ form a probability vector; (A5) $I_g$ grows with $N$ so that $|I_g| \to \infty$.

**Step 1: Setup and notation.** Fix a state $s = (x_\star, \mathcal{M})$ with measured set $\mathcal{M}$ and unmeasured set $\mathcal{U} = \{1, \ldots, M\} \setminus \mathcal{M}$. Define the target for record $i$:
$$g_i = \beta_{\mathcal{M}}^\top \mathbf{y}_{i,\mathcal{M}} + \beta_{\mathcal{U}}^\top \mathbf{y}_{i,\mathcal{U}} + \epsilon_i.$$
The exact conditional variance (under the model) is:
$$\sigma_{\text{cond}}^2(\mathcal{M}) = \beta_{\mathcal{U}}^\top \Sigma_{\mathcal{U}} \beta_{\mathcal{U}} + \sigma_\epsilon^2.$$
The estimator used by VI-Sim at $s$ is:
$$\widehat{\sigma}_{\text{cond}}^2(\mathcal{M}) = \sum_{i \in I_g} \tilde{w}_i(s) \left( g_i - \sum_{j \in I_g} \tilde{w}_j(s) \, g_j \right)^2.$$

**Step 2: Decomposition via independence.** We decompose each target as $g_i = \underbrace{\beta_{\mathcal{M}}^\top \mathbf{y}_{i,\mathcal{M}}}_{\text{measured term}} + \underbrace{z_i}_{\text{unmeasured term}}$, where
$$z_i := \beta_{\mathcal{U}}^\top \mathbf{y}_{i,\mathcal{U}} + \epsilon_i.$$
By assumptions (A1)–(A2) on independence, $z_i$ is independent of $\mathbf{y}_{i,\mathcal{M}}$ and thus independent of any measurable function of $\mathbf{y}_{i,\mathcal{M}}$, including $w_i(s)$ and $\tilde{w}_i(s)$. This yields:
$$\mathbb{E}(z_i \,|\, \mathbf{y}_{i,\mathcal{M}}) = \mathbb{E}(z_i) = \mathbb{E}(\beta_{\mathcal{U}}^\top \mathbf{y}_{i,\mathcal{U}}) + \mathbb{E}(\epsilon_i) = 0, \tag{47}$$
$$\text{Var}(z_i \,|\, \mathbf{y}_{i,\mathcal{M}}) = \text{Var}(z_i) = \text{Var}(\beta_{\mathcal{U}}^\top \mathbf{y}_{i,\mathcal{U}}) + \text{Var}(\epsilon_i) \tag{48}$$
$$= \beta_{\mathcal{U}}^\top \Sigma_{\mathcal{U}} \beta_{\mathcal{U}} + \sigma_\epsilon^2. \tag{49}$$

**Step 3: Analysis of the weighted variance estimator.** Define the weighted means: $\bar{g}_w := \sum_{i \in I_g} \tilde{w}_i(s) g_i$, $\bar{y}_{\mathcal{M},w} := \sum_{i \in I_g} \tilde{w}_i(s) \beta_{\mathcal{M}}^\top \mathbf{y}_{i,\mathcal{M}}$, and $\bar{z}_w := \sum_{i \in I_g} \tilde{w}_i(s) z_i$. Then the variance estimator becomes:
$$\widehat{\sigma}_{\text{cond}}^2(\mathcal{M}) = \sum_{i \in I_g} \tilde{w}_i(s) \left( g_i - \bar{g}_w \right)^2 \tag{50}$$
$$= \sum_{i \in I_g} \tilde{w}_i(s) \left( (\beta_{\mathcal{M}}^\top \mathbf{y}_{i,\mathcal{M}} + z_i) - (\bar{y}_{\mathcal{M},w} + \bar{z}_w) \right)^2 \tag{51}$$
$$= \sum_{i \in I_g} \tilde{w}_i(s) \left( \beta_{\mathcal{M}}^\top \mathbf{y}_{i,\mathcal{M}} - \bar{y}_{\mathcal{M},w} + z_i - \bar{z}_w \right)^2. \tag{52}$$

Taking expectation conditional on the *entire* measured panel $\{\mathbf{y}_{i,\mathcal{M}}\}_{i \in I_g}$ (which determines $\{\tilde{w}_i(s)\}_{i \in I_g}$), and using $\mathbb{E}(z_i|\mathbf{y}_{i,\mathcal{M}}) = 0$ with independence across $i$:
$$\mathbb{E}\left[ \widehat{\sigma}_{\text{cond}}^2(\mathcal{M}) \,\big|\, \{\mathbf{y}_{i,\mathcal{M}}\} \right] = \sum_{i \in I_g} \tilde{w}_i(s) \left( \beta_{\mathcal{M}}^\top \mathbf{y}_{i,\mathcal{M}} - \bar{y}_{\mathcal{M},w} \right)^2 + \sum_{i \in I_g} \tilde{w}_i(s) \, \mathbb{E}\left[ (z_i - \bar{z}_w)^2 \,\big|\, \{\mathbf{y}_{i,\mathcal{M}}\} \right].$$

**Step 4: Simplification of the second term.** The second sum simplifies because $\{z_i\}$ are i.i.d., mean zero and independent of the weights:
$$\mathbb{E}\left[ (z_i - \bar{z}_w)^2 \,\big|\, \{\mathbf{y}_{i,\mathcal{M}}\} \right] = \mathbb{E}[z_i^2] + \mathbb{E}[\bar{z}_w^2] - 2\mathbb{E}[z_i \bar{z}_w] \quad \text{(expanding the square)} \tag{53}$$
$$= \text{Var}(z_i) + \text{Var}(\bar{z}_w) - 2\,\text{Cov}(z_i, \bar{z}_w) \tag{54}$$
$$= \text{Var}(z_i) + \text{Var}(z_i) \sum_{j \in I_g} \tilde{w}_j(s)^2 - 2\tilde{w}_i(s)\text{Var}(z_i) \tag{55}$$
$$= \text{Var}(z_i) \left(1 + \sum_{j \in I_g} \tilde{w}_j(s)^2 - 2\tilde{w}_i(s)\right), \tag{56}$$

where we used $\mathrm{Var}(\bar{z}_w) = \mathrm{Var}(z_i) \sum_{j \in I_g} \tilde{w}_j(s)^2$ (by independence) and $\mathrm{Cov}(z_i, \bar{z}_w) = \tilde{w}_i(s)\, \mathrm{Var}(z_i)$. Therefore

$$\sum_{i \in I_g} \tilde{w}_i(s)\, \mathbb{E}\big[(z_i - \bar{z}_w)^2 \,\big|\, \{\mathbf{y}_{i,\mathcal{M}}\}\big] = \mathrm{Var}(z_i) \sum_{i \in I_g} \tilde{w}_i(s)\, \big(1 + \sum_{j \in I_g} \tilde{w}_j(s)^2 - 2\tilde{w}_i(s)\big) \tag{57}$$

$$= \mathrm{Var}(z_i) \Big( \sum_{i \in I_g} \tilde{w}_i(s) + \sum_{i \in I_g} \tilde{w}_i(s) \sum_{j \in I_g} \tilde{w}_j(s)^2 - 2 \sum_{i \in I_g} \tilde{w}_i(s)^2 \Big) \tag{58}$$

$$= \mathrm{Var}(z_i) \Big( 1 + \sum_{j \in I_g} \tilde{w}_j(s)^2 - 2 \sum_{i \in I_g} \tilde{w}_i(s)^2 \Big) \tag{59}$$

$$= \mathrm{Var}(z_i) \Big( 1 - \sum_{i \in I_g} \tilde{w}_i(s)^2 \Big). \tag{60}$$

Note: In the third line, we used $\sum_{i \in I_g} \tilde{w}_i(s) = 1$ and $\sum_{i \in I_g} \tilde{w}_i(s) \sum_{j \in I_g} \tilde{w}_j(s)^2 = \sum_{j \in I_g} \tilde{w}_j(s)^2$ since the weights sum to 1. Thus

$$\mathbb{E}\big[\widehat{\sigma}^2_{\mathrm{cond}}(\mathcal{M}) \,\big|\, \{\mathbf{y}_{i,\mathcal{M}}\}\big] = \underbrace{\sum_{i \in I_g} \tilde{w}_i(s) \big(\beta_{\mathcal{M}}^\top \mathbf{y}_{i,\mathcal{M}} - \bar{y}_{\mathcal{M},w}\big)^2}_{\text{weighted variance of measured part}} + \mathrm{Var}(z_i) \Big( 1 - \sum_{i \in I_g} \tilde{w}_i(s)^2 \Big).$$

**Asymptotics and conclusion.** By (A5) and boundedness of the weights (since $\sum_i \tilde{w}_i(s) = 1$ and $0 \le \tilde{w}_i(s) \le 1$), we have $\sum_{i \in I_g} \tilde{w}_i(s)^2 \to 0$ in probability when $|I_g| \to \infty$ and the weights are not degenerate.[2] Also, by a (weighted) law of large numbers for triangular arrays with random but *measured-part*-measurable weights and finite second moments, the first term converges in probability to the *true* conditional variance of the measured contribution *given the measured panel*. However, the exact VI-Theo conditional variance *does not depend* on the measured panel (independence across assays), hence

$$\sum_{i \in I_g} \tilde{w}_i(s) \big(\beta_{\mathcal{M}}^\top \mathbf{y}_{i,\mathcal{M}} - \bar{y}_{\mathcal{M},w}\big)^2 \xrightarrow{p} 0.$$

Combining, we get

$$\widehat{\sigma}^2_{\mathrm{cond}}(\mathcal{M}) \xrightarrow{P} \mathrm{Var}(z_i)\,(1 - 0) = \beta_{\mathcal{U}}^\top \Sigma_{\mathcal{U}} \beta_{\mathcal{U}} + \sigma^2_\epsilon = \sigma^2_{\mathrm{cond}}(\mathcal{M}).$$

Hence, under the synthetic linear/independent model, VI-Sim's variance estimator is consistent for the exact VI-Theo conditional variance.

**Implications.** Because $\sigma^2_{\mathrm{cond}}(\mathcal{M})$ is constant in $\mathbf{y}_{\mathcal{M}}$ under independence, any reasonable similarity weighting over measured assays yields the same limiting conditional variance. In more general (correlated or non-linear) settings, the estimator targets $\mathrm{Var}(g \mid \mathbf{y}_{\mathcal{M}} = \text{candidate})$ provided the kernel and bandwidth obey standard nonparametric conditions; IBMDP's stochastic ensembling further mitigates finite-sample bias/variance.

### D.7 EXPERIMENTAL PROTOCOL AND METRICS

**Overview.** We conduct a systematic comparison of the three planning methods across 100 independent trials, focusing on their alignment with the theoretically optimal policy.

For each of 100 independent trials:

(i) Generate a fresh synthetic dataset as in Section D.1.

(ii) Compute VI-Theo's optimal first action at the initial state.

(iii) Compute VI-Sim's recommended first action.

---

[2] Specifically, if $\max_i \tilde{w}_i(s) \xrightarrow{P} 0$, then $\sum_i \tilde{w}_i(s)^2 \le \max_i \tilde{w}_i(s) \sum_i \tilde{w}_i(s) \to 0$. This condition requires that the similarity kernel bandwidth is chosen such that as $N \to \infty$, no single historical case dominates the weights.

(iv) Run IBMDP with an ensemble of MCTS-DPW planners; record (a) the *Top-1* action (most frequent across the ensemble), and (b) the *Top-2* action set (two most frequent).

We report three alignment metrics per trial:

- **T1 Match:** indicator that IBMDP's Top-1 equals VI-Theo's action.

- **T2 Match:** indicator that the VI-Theo action appears in IBMDP's Top-2 set.

- **Sim Match:** indicator that VI-Sim equals VI-Theo.

## D.8    EXPERIMENTAL RESULTS AND ANALYSIS

**Summary.**    Table 5 presents the main results. Over 100 trials, IBMDP's Top-1 matches the VI-Theo optimum in 47 cases; IBMDP's Top-2 covers the optimum in 66 cases; VI-Sim matches the optimum in 36 cases. These results demonstrate that the stochastic, ensemble planner recovers a larger fraction of near-equivalent high-value actions than a deterministic similarity planner, validating the value of IBMDP's ensemble approach.

Table 5: Policy alignment with the theoretical baseline over 100 trials.

| Method | Matches | Match Rate (%) |
|---|---|---|
| IBMDP Top 1 | 47 | 47.0 |
| IBMDP Top 2 | 66 | 66.0 |
| VI-Sim | 36 | 36.0 |

**Statistical Consistency Across Independent Trials.**    To validate the statistical reproducibility of our method, we present the complete trial-by-trial alignment results below. This detailed analysis demonstrates that IBMDP's policy recommendations consistently align with the theoretical optimum across diverse problem instances, providing empirical evidence of the method's robustness and reliability (feature indices refer to assays $a \in \{1, \ldots, 6\}$).

Table 6: Trial-wise comparison of VI-Theo vs. IBMDP and VI-Sim.

| Iter | VI-Theo | IBMDP Top-1 Features | T1 Match | IBMDP Top-2 Features | T2 Match | VI-Sim | Sim Match |
|---|---|---|---|---|---|---|---|
| 1 | 5 | $\{3, 4\}$ | 0 | $\{3, 5\}$ | 1 | 3 | 0 |
| 2 | 5 | $\{3, 4\}$ | 0 | $\{3, 5, 6\}$ | 1 | 3 | 0 |
| 3 | 3 | $\{3, 4\}$ | 1 | $\{3, 5\}$ | 1 | 5 | 0 |
| 4 | 5 | $\{3, 4\}$ | 0 | $\{3, 5\}$ | 1 | 3 | 0 |
| 5 | 3 | $\{3, 4\}$ | 1 | $\{3, 5\}$ | 1 | 3 | 1 |
| 6 | 4 | $\{3, 4\}$ | 1 | $\{3, 5\}$ | 0 | 3 | 0 |
| 7 | 3 | $\{3, 4\}$ | 1 | $\{3, 5\}$ | 1 | 3 | 1 |
| 8 | 5 | $\{3, 4\}$ | 0 | $\{3, 5, 6\}$ | 1 | 3 | 0 |
| 9 | 3 | $\{3, 4\}$ | 1 | $\{3, 5\}$ | 1 | 3 | 1 |
| 10 | 4 | $\{3, 4\}$ | 1 | $\{3, 5, 6\}$ | 0 | 6 | 0 |
| 11 | 4 | $\{3, 4\}$ | 1 | $\{3, 5\}$ | 0 | 4 | 1 |
| 12 | 4 | $\{3, 4\}$ | 1 | $\{3, 5\}$ | 0 | 4 | 1 |
| 13 | 4 | $\{3, 4\}$ | 1 | $\{3, 5\}$ | 0 | 6 | 0 |
| 14 | 4 | $\{3, 4\}$ | 1 | $\{3, 5\}$ | 0 | 6 | 0 |
| 15 | 5 | $\{3, 4\}$ | 0 | $\{3, 5\}$ | 1 | 3 | 0 |
| 16 | 5 | $\{3, 4\}$ | 0 | $\{3, 5\}$ | 1 | 3 | 0 |
| 17 | 4 | $\{3, 4\}$ | 1 | $\{3, 5\}$ | 0 | 4 | 1 |
| 18 | 4 | $\{3, 4\}$ | 1 | $\{3, 5, 6\}$ | 0 | 3 | 0 |
| 19 | 5 | $\{3, 4\}$ | 0 | $\{3, 5\}$ | 1 | 3 | 0 |
| 20 | 5 | $\{3, 4\}$ | 0 | $\{3, 5\}$ | 1 | 3 | 0 |
| 21 | 5 | $\{3, 4\}$ | 0 | $\{3, 5\}$ | 1 | 3 | 0 |
| 22 | 4 | $\{3, 4\}$ | 1 | $\{3, 5\}$ | 0 | 4 | 1 |
| 23 | 5 | $\{3, 4\}$ | 0 | $\{3, 5\}$ | 1 | 3 | 0 |
| 24 | 4 | $\{3, 4\}$ | 1 | $\{3, 5\}$ | 0 | 6 | 0 |
| 25 | 5 | $\{3, 4\}$ | 0 | $\{3, 6\}$ | 0 | 5 | 1 |
| 26 | 5 | $\{3, 4\}$ | 0 | $\{3, 5\}$ | 1 | 3 | 0 |

Table 6: *(continued)*

| Iter | VI-Theo | IBMDP Top-1 Features | T1 Match | IBMDP Top-2 Features | T2 Match | VI-Sim | Sim Match |
|------|---------|----------------------|----------|----------------------|----------|--------|-----------|
| 27 | 5 | {3, 4} | 0 | {3, 5} | 1 | 3 | 0 |
| 28 | 4 | {3, 4} | 1 | {3, 5} | 0 | 3 | 0 |
| 29 | 4 | {3, 4} | 1 | {3, 5} | 0 | 3 | 0 |
| 30 | 5 | {3, 4} | 0 | {3, 5} | 1 | 3 | 0 |
| 31 | 5 | {3, 4} | 0 | {3, 5, 6} | 1 | 3 | 0 |
| 32 | 4 | {3, 4} | 1 | {3, 5} | 0 | 6 | 0 |
| 33 | 5 | {3, 4} | 0 | {3, 5} | 1 | 5 | 1 |
| 34 | 4 | {3, 4} | 1 | {3, 5} | 0 | 4 | 1 |
| 35 | 4 | {3, 4} | 1 | {3, 5} | 0 | 4 | 1 |
| 36 | 5 | {3, 4} | 0 | {3, 5} | 1 | 3 | 0 |
| 37 | 5 | {3, 4} | 0 | {3, 5} | 1 | 3 | 0 |
| 38 | 5 | {3, 4} | 0 | {3, 5} | 1 | 3 | 0 |
| 39 | 5 | {3, 4} | 0 | {3, 5} | 1 | 3 | 0 |
| 40 | 3 | {3, 4} | 1 | {3, 5} | 1 | 3 | 1 |
| 41 | 4 | {3, 4} | 1 | {3, 5} | 0 | 4 | 1 |
| 42 | 4 | {3, 4, 5} | 1 | {3, 4, 5} | 1 | 6 | 0 |
| 43 | 3 | {3, 4} | 1 | {3, 5} | 1 | 3 | 1 |
| 44 | 5 | {3, 4} | 0 | {3, 5, 6} | 1 | 3 | 0 |
| 45 | 5 | {3, 4} | 0 | {3, 5} | 1 | 3 | 0 |
| 46 | 5 | {3, 4} | 0 | {3, 5} | 1 | 3 | 0 |
| 47 | 3 | {3, 4} | 1 | {3, 5} | 1 | 3 | 1 |
| 48 | 5 | {3, 4} | 0 | {3, 5} | 1 | 5 | 1 |
| 49 | 4 | {3, 4} | 1 | {3, 5} | 0 | 3 | 0 |
| 50 | 5 | {3, 4} | 0 | {3, 5} | 1 | 3 | 0 |
| 51 | 4 | {3, 4} | 1 | {3, 5, 6} | 0 | 4 | 1 |
| 52 | 5 | {3, 4} | 0 | {3, 5} | 1 | 3 | 0 |
| 53 | 5 | {3, 4} | 0 | {3, 5} | 1 | 5 | 1 |
| 54 | 5 | {3, 4} | 0 | {3, 5} | 1 | 3 | 0 |
| 55 | 4 | {3, 4} | 1 | {3, 5} | 0 | 3 | 0 |
| 56 | 5 | {3, 4} | 0 | {3, 5} | 1 | 3 | 0 |
| 57 | 4 | {3, 4} | 1 | {3, 4, 5} | 1 | 3 | 0 |
| 58 | 3 | {3, 4} | 1 | {3, 5} | 1 | 3 | 1 |
| 59 | 4 | {3, 4} | 1 | {3, 5} | 0 | 6 | 0 |
| 60 | 3 | {3, 4} | 1 | {3, 4, 5} | 1 | 3 | 1 |
| 61 | 3 | {3, 4} | 1 | {3, 5, 6} | 1 | 3 | 1 |
| 62 | 6 | {3, 4} | 0 | {3, 5} | 0 | 3 | 0 |
| 63 | 5 | {3, 4} | 0 | {3, 5} | 1 | 5 | 1 |
| 64 | 4 | {3, 4} | 1 | {3, 5} | 0 | 3 | 0 |
| 65 | 5 | {3, 4} | 0 | {3, 5} | 1 | 3 | 0 |
| 66 | 5 | {3, 4} | 0 | {3, 5} | 1 | 3 | 0 |
| 67 | 5 | {3, 4} | 0 | {3, 5} | 1 | 3 | 0 |
| 68 | 4 | {3, 4} | 1 | {3, 5} | 0 | 6 | 0 |
| 69 | 6 | {3, 4} | 0 | {3, 5} | 0 | 6 | 1 |
| 70 | 4 | {3, 4} | 1 | {3, 5} | 0 | 4 | 1 |
| 71 | 5 | {3, 4} | 0 | {3, 5} | 1 | 3 | 0 |
| 72 | 5 | {3, 4} | 0 | {3, 5} | 1 | 3 | 0 |
| 73 | 4 | {3, 4} | 1 | {3, 5} | 0 | 6 | 0 |
| 74 | 5 | {3, 4} | 0 | {3, 5} | 1 | 3 | 0 |
| 75 | 5 | {3, 4} | 0 | {3, 5} | 1 | 5 | 1 |
| 76 | 5 | {3, 4} | 0 | {3, 5} | 1 | 5 | 1 |
| 77 | 5 | {3, 4} | 0 | {3, 5} | 1 | 3 | 0 |
| 78 | 4 | {3, 4} | 1 | {3, 5} | 0 | 4 | 1 |
| 79 | 5 | {3, 4} | 0 | {3, 5} | 1 | 3 | 0 |
| 80 | 5 | {3, 4} | 0 | {3, 5} | 1 | 5 | 1 |
| 81 | 5 | {3, 4} | 0 | {3, 5} | 1 | 3 | 0 |
| 82 | 5 | {3, 4} | 0 | {3, 5} | 1 | 3 | 0 |
| 83 | 4 | {3, 4} | 1 | {3, 5} | 0 | 4 | 1 |
| 84 | 5 | {3, 4} | 0 | {3, 5} | 1 | 3 | 0 |
| 85 | 5 | {3, 4} | 0 | {3, 5} | 1 | 3 | 0 |
| 86 | 5 | {3, 4} | 0 | {3, 5} | 1 | 3 | 0 |
| 87 | 4 | {3, 4} | 1 | {3, 5, 6} | 0 | 3 | 0 |
| 88 | 5 | {3, 4} | 0 | {3, 5} | 1 | 3 | 0 |

*Continued on next page*

Table 6: *(continued)*

| Iter | VI-Theo | IBMDP Top-1 Features | T1 Match | IBMDP Top-2 Features | T2 Match | VI-Sim | Sim Match |
|------|---------|----------------------|----------|----------------------|----------|--------|-----------|
| 89   | 5       | $\{3, 4\}$           | 0        | $\{3, 5\}$           | 1        | 3      | 0         |
| 90   | 5       | $\{3, 4\}$           | 0        | $\{3, 5\}$           | 1        | 3      | 0         |
| 91   | 3       | $\{3, 4\}$           | 1        | $\{3, 5\}$           | 1        | 3      | 1         |
| 92   | 4       | $\{3, 4\}$           | 1        | $\{3, 5\}$           | 0        | 4      | 1         |
| 93   | 3       | $\{3, 4\}$           | 1        | $\{3, 5\}$           | 1        | 3      | 1         |
| 94   | 3       | $\{3, 4\}$           | 1        | $\{3, 5\}$           | 1        | 3      | 1         |
| 95   | 4       | $\{3, 4\}$           | 1        | $\{3, 5, 6\}$        | 0        | 3      | 0         |
| 96   | 5       | $\{3, 4\}$           | 0        | $\{3, 5\}$           | 1        | 5      | 1         |
| 97   | 5       | $\{3, 4\}$           | 0        | $\{3, 5\}$           | 1        | 3      | 0         |
| 98   | 4       | $\{3, 4\}$           | 1        | $\{3, 5\}$           | 0        | 4      | 1         |
| 99   | 3       | $\{3, 4\}$           | 1        | $\{3, 5\}$           | 1        | 3      | 1         |
| 100  | 4       | $\{3, 4\}$           | 1        | $\{3, 5\}$           | 0        | 6      | 0         |

**Interpretation.** VI-Theo and VI-Sim return a single deterministic action per state. IBMDP explores the posterior-predictive policy space via stochastic rollouts and, by ensembling, surfaces *multiple* near-equivalent high-value choices. The superior Top-2 coverage (66% vs. VI-Sim's 36% matching rate) reflects better policy-space exploration and robustness to finite-sample effects.

# E  BENCHMARK WITH PUBLIC DATASET

## E.1  HIGH-COST DIFFERENTIAL CLEARANCE OPTIMIZATION

We reuse a publicly available pharmacokinetics dataset (rat, dog, human clearance plus QSAR predictors) to stress-test IBMDP under large assay cost differentials. The dataset is described in (**?**) and is available to download. The planner may propose at most two assays per decision step, and the expensive human clearance assay is treated just like the rat and dog assays (i.e., it can be scheduled in any batch). The operational objective is to finish with human clearance exceeding 1.0 mL/min/kg while spending as little as possible. Species-specific costs are listed in Table 7.

| Assay | Cost ($) | Relative Cost |
|-------|----------|---------------|
| Rat clearance | 400 | 1.0× |
| Dog clearance | 800 | 2.0× |
| Human clearance | 4,000 | 10.0× |

Table 7: Assay cost structure for high-cost differential experiment

Unlike traditional gated progression (e.g., "rat before dog before human"), every episode starts with the unmeasured state $s_0 = \{\text{CL}_{\text{rat}}^{\text{pred}}, \text{CL}_{\text{dog}}^{\text{pred}}, \text{CL}_{\text{human}}^{\text{pred}}\}$ so the solver can pick any eligible batch. The IBMDP ensemble (30 runs, $c = 5.0$, 5,000 iterations per run, $\tau \in \{0.6, 0.9\}$) produces a Maximum-Likelihood Action-Set Path (MLASP) by majority vote over recommended assay batches. The voting tally reveals three regimes: (i) high-uncertainty states favour rat/dog assays before committing to human tests; (ii) low-uncertainty states jump directly to human clearance; and (iii) intermediate states switch behaviour depending on the belief threshold $\tau$. Figure 5 visualizes the resulting Pareto front and highlights how the MLASP navigates the trade-off between total spend and terminal uncertainty.

## E.2  INTERPRETING THE PARETO FRONTIER

Figure 5 aggregates planning outcomes for tolerances $\tau \in \{0.0, 0.1, \ldots, 1.0\}$ under the two-assays-per-step constraint. For each tolerance we execute a 30-run ensemble and record the first assay batch proposed by every run. Each marker therefore represents the rule "if $H(s_T) \leq \tau$ is required, begin with batch $A_0$"; the horizontal axis reports the corresponding assay spend (rat + dog + human) and the vertical axis equals the targeted uncertainty $\tau$. The blue locus links the Pareto-efficient points, exposing the spend-versus-uncertainty trade-off that emerges when $\tau$ is tightened. The starred marker denotes the Maximum-Likelihood Action-Set Path (MLASP)—the batch occurring most frequently across ensemble members for the displayed tolerance. Progressing from higher to lower $\tau$ shows that lenient tolerances favour inexpensive rat/dog assays, whereas stringent requirements such as $H(s_T) \leq 0.10$ eventually demand the human clearance assay despite its $10\times$ cost in Table 7. After

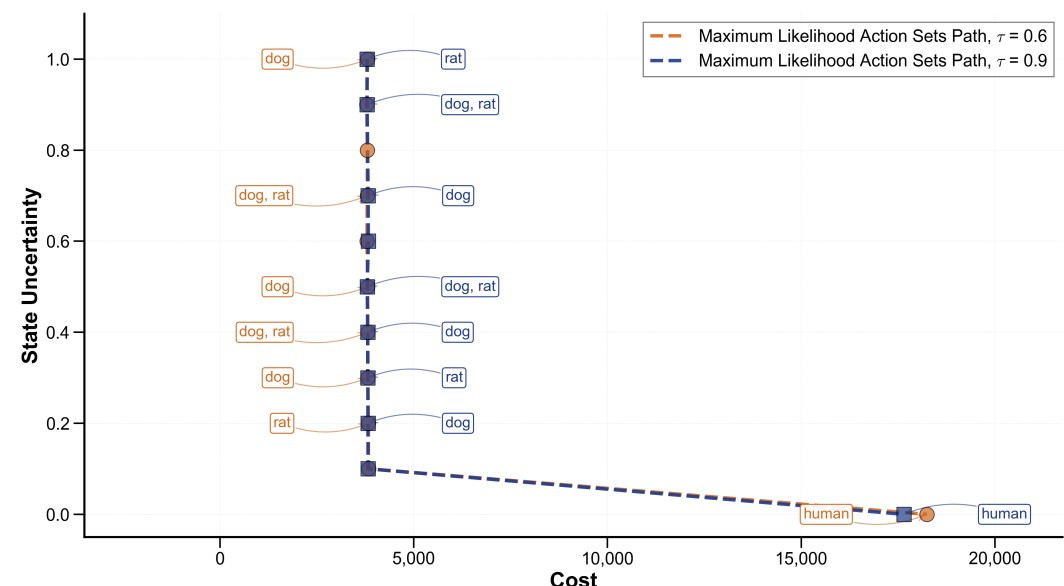

Figure 5: ADME clearance optimization results comparing IBMDP performance under two belief thresholds ($\tau = 0.6$ and $\tau = 0.9$) for a representative compound from the public CNS clearance benchmark. The plot demonstrates the Pareto-optimal trade-offs between total assay spend (horizontal axis) and terminal state uncertainty $H(s_T)$ (vertical axis) achieved by the IBMDP ensemble across 30 runs. The two distinct curves for $\tau = 0.6$ (more lenient) and $\tau = 0.9$ (more stringent) illustrate how tighter belief thresholds drive higher assay expenditure to achieve lower uncertainty. Notably, the two tau configurations exhibit strong alignment in their Pareto frontiers, confirming that IBMDP produces consistent and robust planning strategies across different confidence requirements. The Maximum-Likelihood Action-Set Paths (MLASPs) for each threshold are marked, showing how the ensemble consensus adapts to balance the high cost of human clearance assays ($4,000) against the need to reduce decision uncertainty below the specified threshold.

the first batch is executed the IBMDP policy updates the belief state and recomputes the next action, so the figure captures the initial decision while the full policy remains adaptive.

## F    USE OF LLM

We used a large language model (LLM) solely as a general-purpose writing aid for light copyediting and polishing. Specifically, the LLM was used to improve grammar, clarity, and flow of sentences written by the authors, and to suggest minor phrasing alternatives. The LLM did not contribute to research ideation, methodology, experimental design, data analysis, interpretation of results, or substantive content generation. All technical claims, analyses, references, and conclusions were conceived, written, and verified by the authors. The authors take full responsibility for all content in this paper, including any text that was edited with the assistance of an LLM. No LLM is listed as an author, and no text was accepted without author review and verification.

## G    GLOBAL NOTATION REFERENCE

This appendix provides a comprehensive reference for all mathematical notation used throughout the manuscript. The table below organizes symbols by category for easy reference.

Table 8: Global Notation Reference summarizing the symbols used across the manuscript.

| Symbol | Meaning |
|---|---|
| **Sets & Indices** | |
| $N, M$ | Number of historical compounds and total available assays, respectively. |
| $X = \{x_i\}_{i=1}^N$ | Set of $N$ historical compounds with fixed representations. |
| $\mathcal{A} = \{a_1, \ldots, a_M\}$ | Set of $M$ available assays. |
| $\mathcal{D} = \{(x_i, \mathbf{y}_i)\}_{i=1}^N$ | Historical dataset of compounds and their assay outcome vectors. |
| $i, j, k, t$ | Indices for historical case, assay, feature, and decision step. |
| **Candidate Compound & State** | |
| $x_\star$ | The candidate compound for which a plan is being made. |
| $s_t = (x_\star, \{y_{\star,j}\}_{j \in M_t})$ | State at step $t$, comprising the candidate and all outcomes measured so far. |
| $M_t \subseteq \mathcal{A}$ | The set of assays that have been **m**easured for $x_\star$ up to step $t$. |
| $U_t = \mathcal{A} \setminus M_t$ | The set of **u**nmeasured assays for $x_\star$ at step $t$. |
| **Actions, Costs & Policy** | |
| $\mathcal{A}_t = \mathcal{P}_{\leq m}(U_t) \cup \{\text{eox}\}$ | Action set at $s_t$: batches of up to $m$ unmeasured assays, plus the stop action. |
| $m$ | Maximum number of assays that can be run in parallel per step. |
| $A_t \in \mathcal{A}_t$ | The action (a batch of assays) chosen at step $t$. |
| $c(s_t, A_t) \in \mathbb{R}_{\geq 0}^q$ | Vector of $q$ resource costs for taking action $A_t$. |
| $\boldsymbol{\rho} \in \mathbb{R}_{\geq 0}^q$ | User-defined weights for trading off different cost types. |
| $R(s_t, A_t)$ | Scalar step cost: $\boldsymbol{\rho}^\mathsf{T} c(s_t, A_t)$. $R(s_t, \text{eox}) = 0$. |
| $\pi, \pi^\star$ | A policy mapping states to actions, and the optimal policy. |
| **Similarity Model & Target Functionals** | |
| $g$ | The primary scalar target property of interest (e.g., an *in vivo* endpoint). |
| $G = \{g_i\}, I_g$ | Set of historical target values and the index set where they are available. |
| $d(s_t, D_i)$ | Variance-normalized distance between the current state and historical case $i$. |
| $w_i(s_t)$ | Similarity weight of historical case $i$ given the current state $s_t$. |
| $\tilde{w}_i(s_t)$ | Similarity weight $w_i(s_t)$ re-normalized over the set $I_g$. |
| $H(s_t)$ | State uncertainty: the weighted variance of the target $g$ based on $\tilde{w}_i(s_t)$. |
| $L(s_t)$ | Goal likelihood: the weighted probability that $g$ is in a desirable range. |
| $\mathbf{1}[\cdot]$ | Indicator function: returns 1 when the condition inside the brackets holds, and 0 otherwise. |
| **Hyperparameters & Constraints** | |
| $\lambda_w, \lambda_k$ | Hyperparameters: global similarity bandwidth and per-feature weights. |
| $\epsilon, \tau$ | Thresholds for the constrained objective: max terminal uncertainty and min goal likelihood. |
| $\gamma, T$ | Discount factor and maximum horizon for the MDP. |
| $N_e, n_{\text{itr}}$ | Planning parameters: ensemble size and MCTS iterations per run. |
| **Algorithm Components** | |
| MCTS-DPW | Monte Carlo Tree Search with Double Progressive Widening. |
| MLASP | Maximum-Likelihood Action-Sets Path: final plan from ensemble majority voting. |
| eox | End of experiment action (stop action). |

