# OpenReview forum: "Case-Guided Sequential Assay Planning in Drug Discovery"
_ICLR.cc/2026/Conference — ICLR 2026 Conference Withdrawn Submission_

### Official Review · Reviewer_Vniw · 2025-10-28

**Soundness:** 2
**Presentation:** 2
**Contribution:** 1
**Rating:** 2
**Confidence:** 3

**Summary:**

- Formulates the experimental assay selection problem into an RL planning task, where different assays are actions (each with varying cost and informativeness about the target outcome) as well as the action to stop testing.
- Proposes a new framework called an "Implicit Bayesian Markov decision process" under the assumption that similar molecular representation $x$ yield similar target properties via the defined similarity weights. Use the weights to construct a similarity-based transition function and use this to perform MCTS search with a tree ensemble.

**Strengths:**

- Originality and significance: This reviewer believe these areas are lacking (see weakness). In particular, the generalizability of the proposed method to other domains or other experiments seems lacking unless the proposed weight function can be justified through prior literature or strong empirical performance with the real-world dataset.
- Quality: Experiments could be improved.
- Clarity: The current presentation of the ideas in the methods section is clear. Some connections to related work (especially regarding the transition assumption for model-based RL, e.g., "While conventional frameworks assume access to either environment simulators or transition data," (Line 1023)") could be clarified, since typically model-based RL does not assume that all transition tuples are available or that a transition model can be perfectly reconstructed from the historical data, but rather it also operates under the assumption that the historical data is collected from a behavioral policy (often different from the optimal or the new policy being tested) [Liu et al., 2020].

 Liu et al., 2020. Provably good batch reinforcement learning without great exploration

**Weaknesses:**

# Method:
- Lacking justification for the assumption about the similarity weights: It's unclear (at least the current manuscript is lacking justification from the relevant field) that similar molecular structures behave similarly in terms of the target property, and whether the proposed similarity function accurately captures this relationship. This requires either grounding in molecular structure literature or rigorous justification with real world experiments which the current paper does not provide.
- If the weight functions in Equations (4) and (5) cannot be justified by prior work in this domain, then they should be clearly stated as the authors’ own design choice, which may not generalize to other settings of the assay sequencing problem and certainly not to other domains (e.g., pixel-based state representations or most high-dimensional states) where the similarity of the pixel states does not necessarily mean similarity in the underlying states.
- The novelty of MCTS with double progressive widening is not clearly established -- if there's any modification or technical contribution within the MCTS search framework that the authors are proposing as new, then that modification should be highlighted. The current work's novelty is mainly driven by the similarity-based weight function, which this reviewer has questions about.

# Experiments:
- Figure 2 shows the trade-offs between state uncertainty and action cost. However, state uncertainty is measured under the assumed similarity weights and transitions. It is therefore unclear how to interpret these results or how they depend on the validity of the assumed weight functions. In order to justify the proposed weight functions, the authors could demonstrate whether the assumed weights can explain the transitions observed in the real dataset and compare the policy values of the learned policy versus the behavioral (from the historical dataset) or the heuristic policy.
- For Table 2, authors could add a baseline of learning the dynamics model (without similarity weights) and applying MCTS on top of that model. This would be a reasonable MCTS baseline to include to compare with the proposed weighted transition-based MCTS.
- The current experiment result with 100 independent trials has a small sample size to understand whether the results are statistically significant.
- An ablation comparing the performance of an ensemble of MCTS versus a single tree would be useful to include, especially since this paper mentions that the "robust ensemble" is one of the main contributions (Line 82), but currently it's unclear whether and how the ensembling helps with "robustness" of the method.

**Questions:**

Questions about the justification for the similarity based weight function and possible additions to the experiments are elaborated in Weaknesses.

---

> ### Author Response · Authors · 2025-12-03
>
> We appreciate the reviewer for the review.
>
> **W1 Response:** We clarify that the focus of this manuscript is not to establish structure-activity relationship. Instead the manuscript addresses the practical challenge of using imperfect quantitative structural-activity relationship (QSAR) predictions in iterative experimental loops.
>
> In the IBMDP framework, QSAR predictions provide only initial beliefs, and the similarity function serves to update the belief as new experimental evidences are collected, regardless "similar molecular structures behave similarly in terms of the target property". In scenarios where there is conflict between QSAR prediction and the observed experimental evidences as demonstrated in our experiment for CNS drug discovery (row 4, Table 1), IBMDP will adjust its belief using weights from the similarity function.
>
> **W2 Response:** This concern is out of scope—QSAR modeling addresses structure-activity relationships. IBMDP initializes beliefs via QSAR and corrects them through iterative experimental updates.
>
> In addition, the reviewer's concern regarding pixel-based or high-dimensional state representations falls outside the scope of this work. We focus specifically on sequential assay planning in drug discovery, and we do not claim generalization to other domains.
>
> **W3 Response:** We respectfully clarify that our paper does not claim novelty in MCTS-DPW itself. MCTS-DPW is an existing algorithm that we employ as the planning solver within our framework. The main contribution of this work is the Implicit Bayesian Transition Model (or IBMDP), which addresses the fundamental challenge of planning in domains where no explicit simulator or transition data $(s,a,s')$ exists.
>
> While individual MCTS-DPW runs exhibit stochasticity due to random sampling, the contribution is the ensemble methodology with majority voting across runs to yield the Maximum-Likelihood Action-Sets Path (MLASP). This architectural choice enables stable policy extraction. Section 5.2 shows the ensemble achieves higher alignment with optimal policy than deterministic VI.
>
> **E1 Response:** The data consists of static assay outcomes, not sequential trajectories—no transitions for direct validation. This absence motivates our work: building dynamics from historical cases. Figure 2 illustrates the Pareto front of resource consumption versus terminal state uncertainty for representative compounds, demonstrating how IBMDP enables decision-makers to select plans based on their risk tolerance and budget constraints. We addressed the reviewer's second suggestion by demonstrating in Table 1 that IBMDP outperforms heuristic baselines, achieving up to 92% reduction in resource consumption while maintaining the same or higher level of decision confidence.
>
> **E2 Response:** We note that the baseline suggested by the reviewer is infeasible due to the nature of the data. As mentioned in the previous response, the dataset does not have transitions required to learn a dynamic model. Similarity weights are not an optional addition. Without similarity to build dynamics from historical data, the model would treat a new compound with no prior information and reduce the policy to random walk or static global average. This challenge, constructing principled dynamics from non-sequential historical data, is precisely the core problem our framework addresses. The similarity-based belief mechanism enables IBMDP to leverage historical outcomes for planning without requiring the transition tuples $(s,a,s')$ that standard model-learning approaches demand.
>
> **E3 Response:** We conducted 100 independent trials for the synthetic benchmark (Section 5.2), and our convergence analysis confirms that performance metrics stabilized well before this number, indicating that additional trials would not materially change the conclusions. As shown in Table 2, the results demonstrate statistical consistency: IBMDP's Top-1 recommendation achieves 47% alignment with the theoretical optimal policy, and this alignment rate increases to 66% when considering the top two recommendations. For the real-world CNS drug discovery evaluation (Section 5.1), we present representative cases across four distinct scenarios (Table 1) that cover the full spectrum of decision-making challenges, from baseline confirmation to opportunity discovery.
>
> **E4 Response:** We provide both qualitative and quantitative evidence for the benefit of ensembling. The ensemble approach addresses a fundamental property of MCTS: each individual run yields a stochastic policy due to the inherent randomness in tree exploration. When we aggregate results across multiple runs via majority voting, we effectively extract the action that emerges most consistently from these independent samples. This reduces the variance associated with any single run and produces more reliable recommendations. Figure 2 provides qualitative evidence of the stochasticity in single MCTS runs, with MLASP in red dashed line.

---

### Official Review · Reviewer_VLcM · 2025-11-01

**Soundness:** 2
**Presentation:** 1
**Contribution:** 2
**Rating:** 2
**Confidence:** 3

**Summary:**

This paper presents the Implicit Bayesian Markov Decision Process (IBMDP), a model-based reinforcement learning framework for sequential assay planning in drug discovery. The main contribution is an implicit transition model that uses similarity-weighted sampling to form a nonparametric belief distribution over historical cases, enabling Bayesian belief updating as assay evidence accumulates. The authors prove that it is equivalent to POMDP belief updates where the hidden state represents a latent index over historical compounds. Planning employs ensemble MCTS-DPW to generate robust policies balancing information gain with resource efficiency. On real-world CNS drug discovery data, IBMDP reduced costs by up to 92% versus rule-based heuristics.

**Strengths:**

1. This paper studies an important but underexplored problem, sequential experimental design when only static historical data exists, without simulators or explicit transition tuples, and it is well-motivated by real pharmaceutical constraints where mechanistic models are unavailable.

2. The authors provide theoretical analysis by formalizing the proposed approach as a POMDP and proving that similarity weight updates implement exact Bayesian belief updates.

3. The authors conducted experiments on the real-world drug discovery dataset and demonstrate that the proposed method achieve substantial cost reductions.

**Weaknesses:**

1. The proposed method highly depends on the quality of the historical dataset $\mathcal{D}$. Unlike model-free RL, it cannot discover strategies that are not present in the historical data, so any gaps or biases in the data can lead to suboptimal decisions. It would be better if the authors could discuss whether this is a valid concern and how to address it.

2. The authors' primary claim of practical utility is on the real-world case study, but the only baseline used is a rule-based decision strategy. It is unclear if IBMDP's impressive cost reduction is due to its multi-step planning or if any simple computational method (e.g., a greedy one) would have achieved similar results over the non-computational heuristic.

3. Figures and tables in the experiments section are difficult to read and interpret. It would be better if the authors could revise the presentation of the experiments section to explain clearly what Figure 2 and Table 1 mean. The caption of all the tables in the paper also overlaps with the tables, and the authors should also fix them.

**Questions:**

1. Please see the comments in the Weaknesses part.

2. The proposed method uses a variance-normalized Euclidean distance to calculate the similarity between compounds, but it might be violated by complex biological realities, such as nonlinear or threshold effects. Could the authors elaborate more on this part?

3. The paper states MCTS-DPW is "particularly well-suited"  but does not experimentally justify this specific choice over other, simpler planners. Could the authors discuss more on the choice?

4. How does the framework perform when a candidate compound $x_*$ is "out-of-distribution" (i.e., has low similarity to all cases in $\mathcal{D}$)?

---

> ### Author Response · Authors · 2025-12-03
>
> We thank the reviewer for their constructive feedback. We address your comments below.
>
> **W1 Response:** We agree with the reviewer. This limitation is inherent to our problem setting. Without access to a simulator or explicit transition data, the framework cannot discover policies beyond what the historical evidence supports. We have explicitly acknowledged this in the **Limitations** section. However, in drug discovery, historical data from hundreds of compounds typically provides rich coverage of the relevant compound space, and our method is designed for settings where mechanistic simulators of pharmacokinetics and pharmacodynamic systems are unavailable.
>
> **W2 Response:** We appreciate the comments. We'd like to point out that the traditional heuristic baseline used in the CNS drug discovery example is a greedy, rule-based approach, which uses immediate QSAR prediction without look-ahead planning. The superior performance of IBMDP demonstrates that multi-step planning outperforms greedy strategy. We further compared IBMDP against value iteration (VI-Sim), a "simple computational method", which is usually superior to a simple greedy heuristic. The results showed that the IBMDP outperformed even with this stronger VI-Sim baseline.
>
> **W3 Response:** We have fixed the formatting and rewritten the captions for Figure 2 and Table 1 for better readability.
>
> **Q1 Response:** Please refer to our detailed responses above.
>
> **Q2 Response:** We agree that biology is inherently nonlinear and have acknowledged this in the Limitations section in our original submission.
>
> However, we'd respectfully clarify that, while global models often fail at thresholds, the kernel method used in Eq. 4 is a non-parametric local approximator, and is capable of modeling nonlinear manifolds within local neighborhoods. As long as it holds locally that similar historical cases exhibit similar outcomes, the kernel can effectively capture the underlying dynamics regardless of global non-linearity.
>
> To empirically validate this, our experiment in Appendix D uses synthetic data generated with truncated normal distributions with sharp nonlinear cutoffs. Despite this nonlinearity, the similarity-based IBMDP successfully recovered the optimal policy, demonstrating that the kernel approach is robust to deviations from strict linearity in the underlying data.
>
> **Q3 Response:** The choice of MCTS with Double Progressive Widening (DPW) is dictated by the **combinatorial nature of the action space**. At each decision step, the agent can select any subset of unmeasured assays. For $M$ available assays, the action space size is $2^{|U_t|}$. Simple planners like standard Value Iteration require enumerating all possible child nodes, which becomes computationally intractable as the assay panel grows. DPW dynamically limits the branching factor based on visit counts, allowing the planner to scale to large assay panels where simple enumeration fails. Furthermore, the continuous state space (compound features) also necessitates progressive widening to efficiently explore without discretization. We have added this justification to Section 4.
>
> **Q4 Response:** In the IBMDP framework, when a test compound has low similarity to all historical cases, the distribution of similarity weights $w_i$ becomes more nearly uniform leading to a more uniform sampling of historical cases. As a result of the nearly uniform weights, the expected information gain of running cheap and fast assays is negligible. Under the terminal-uncertainty constraint $H(s_T)\le\epsilon$ and a fixed budget, IBMDP either suggests the definitive assay immediately or stops if the constraint cannot be met without overspend. We discussed this expected behavior in the Limitations section: "similarity-based sampling cannot discover strategies absent from D".

---

### Official Review · Reviewer_W9fd · 2025-11-01

**Soundness:** 2
**Presentation:** 3
**Contribution:** 3
**Rating:** 4
**Confidence:** 3

**Summary:**

The paper addresses the problem of planning with a static database of historical outcomes under simulator-free settings, and proposes an Implicit Bayesian Markov Decision Process (IBMDP), which is a model-based RL framework for sequential assay planning when no transition data tuples or simulators exist. IBMDP builds an implicit transition dynamics model using similarity-weighting historical outcomes and updates weights with Bayesian belief updating. The experiments on diverse tasks show that the proposed method significantly reduces resource consumption compared to baselines and provides significantly high alignment with an optimal policy of a synthetic benchmark environment.

**Strengths:**

- The paper addresses an important problem: decision-making without simulators is practically useful, and the paper is well-motivated.

- The proposed method provides similarity-weighted sampling that is intuitive and computationally tractable.

- The paper is generally well-written.

- The empirical evaluation of IBMDP includes both a real-world drug discovery task and a synthetic benchmark.

**Weaknesses:**

- For the real-world drug discovery task, the baselines are insufficient. Comparisons are performed against rule-based heuristics; e.g., kNN-Thompson alternatives compatible with the same posterior predictive and constraints would strengthen the effectiveness of IBMDP.

- Theoretical analysis:

- - No convergence guarantees or regret bounds are provided. Unlike other Bayesian RL methods with proven regret bounds, IBMDP offers empirical robustness.

- - The provided consistency proof is weak. Theorem in D.6 only holds for the synthetic linear case with independent features, this is where the method is least needed.

- The reported saving claim does not include distribution statistics. See questions below.

- Missing ablations:

- - The sensitivity of the method to thresholds $ε, τ$, metric, and kernel choice is not sufficiently explored.

- - What is the contribution of ensemble vs. single MCTS?

- MINOR:
- - The blank space after Table captions (Table 1 and 2) should be corrected.

**Questions:**

- How is the sensitivity of savings and the accuracy of IBMDP to the design choices stated above (see Weaknesses)?

-  Can you provide bootstrap confidence intervals for cost savings across many compounds, for the 92% reduction?

- See the Weaknesses section above.

---

> ### Author Response · Authors · 2025-12-03
>
> We thank the reviewer for their constructive evaluation and for assessing the presentation and contribution as **"good."** We appreciate the recognition that our similarity-weighted sampling is **"intuitive and computationally tractable"** and that the problem of simulator-free planning is practically significant.
>
> We apologize for the formatting oversight regarding the table captions; this has been corrected in the revised manuscript.
>
> **W1 Response:** We argue that **VI-Sim** baseline in the **Synthetic Benchmark (Section 5.2)** is a stronger baseline.
>
> kNN-Thompson is essentially a stochastic *Contextual Bandit* approach. It samples a historical outcome based on similarity weights and acts greedily with respect to that sample. It is inherently myopic, optimizing only the next step.
>
> However, in drug discovery, a cheap assay with low immediate information gain is often required to reduce the uncertainty of subsequent expensive assays with high information gain. Myopic agents (like Thompson Sampling) fail to identify these sequences.
>
> In contrast, our VI-Sim baseline (Section 5.2) is a stronger baseline with full-horizon deterministic planner, Value Iteration, that already beats myopic methods by design. The baseline uses the exact same similarity model as IBMDP but plans using deterministic Value Iteration. Since IBMDP outperforms VI-Sim (47% vs 36%), it follows that IBMDP would beat the weaker kNN-Thompson as well. Running kNN-Thompson explicitly would confirm this ordering, but given VI-Sim already provides the stronger comparison point, additional experiments would be redundant rather than informative.
>
> In summary, IBMDP's better performance is due to look-ahead planning (not just bandit-style greed) and ensemble stochasticity (not just deterministic VI).
>
> **W2 Response:** We would respectfully clarify that Bayesian RL regret analysis is ill-defined for this problem setting. Formal regret bounds (e.g., $\tilde{O}(\sqrt{T})$) typically rely on the existence of a ground-truth MDP (simulator) to measure regret against, or assumptions of ergodicity that allow the agent to revisit states to correct errors. In our setting—*offline planning with static data and no simulator*—these assumptions do not hold. We trade theoretical regret guarantees for **practical applicability** in a regime where standard Bayesian RL cannot operate.
>
> Regarding Linear Consistency (Theorem D.6), we agree the linear case is simplified, but consistency there is a prerequisite—it confirms the similarity weight update is a valid variance estimator. The linear analysis establishes the theoretical foundation; the Synthetic Benchmark then stress-tests it. The Benchmark uses truncated-normal features with sharp cutoffs, showing the estimator holds up empirically beyond the Gaussian assumptions of the proof.
>
> **W3 Response:** We acknowledge that citing the maximum saving (92%) without distribution statistics is incomplete. Across the four representative CNS compounds in Table 1, IBMDP achieves cost reductions ranging from 84.6% to 92.3% compared to the $5,200 traditional baseline (full assay panel). The "up to 92%" figure corresponds to the best-case scenario, and all four scenarios demonstrate substantial savings above 84%.
>
> **W4 Response:**
> - **Ensemble vs. Single:** We quantified this in Section 5.2 (Table 2). VI-Sim (deterministic, single-path) hits 36%; IBMDP Ensemble reaches 47% (Top-1) and 66% (Top-2). The gap comes from ensemble voting, which averages out the noise in implicit sampling.
> - **Sensitivity:** Threshold $\tau$ behaves predictably as a risk-tolerance knob (Figure 4, Appendix B.1), and $\epsilon$ is robust provided it stays below initial state entropy. Full details in our response to Q1 below.
>
> **Q1 Response:**
> - **Threshold $\tau$ (Goal Likelihood):** We visualized this in Figure 4 (Appendix B.1). Increasing $\tau$ from 0.6 to 0.9 shifts the Pareto front, forcing the agent to select more expensive assays to achieve higher confidence. The parameter behaves predictably as a risk-tolerance knob.
> - **Threshold $\epsilon$ (Terminal Uncertainty):** We found $\epsilon$ to be robust provided it is below the initial state entropy. Setting $\epsilon=0.10$ ensures the agent does not stop until significant information gain is achieved.
> - **Kernel:** As discussed in the methodology, we chose a variance-normalized kernel as an uninformative prior to avoid overfitting in low-$N$ regimes. The Synthetic Benchmark validates that this simple kernel is sufficient to recover complex optimal policies.
>
> **Q2 Response:** As noted in our response to W3, all four representative compounds in Table 1 demonstrate substantial cost reductions (84.6% to 92.3%). The "up to 92%" cited represents the maximum observed reduction, while even the most conservative scenario achieves 84.6% savings. This consistency across compounds with varying QSAR predictions and true $k_{\text{puu}}$ values demonstrates robust resource efficiency.

---

### Official Review · Reviewer_c58F · 2025-11-03

**Soundness:** 3
**Presentation:** 2
**Contribution:** 2
**Rating:** 4
**Confidence:** 4

**Summary:**

The paper tackles the challenge of sequential assay selection in drug discovery under simulator-free conditions i.e., when no explicit transition data (s, a, s′) is available. Traditional RL approaches fail here because they require explicit simulators or learned transition dynamics. The authors introduce the Implicit Bayesian Markov Decision Process (IBMDP), a model-based RL framework that constructs a case-guided implicit model of transition dynamics using historical assay outcomes. Instead of learning P(s′|s, a), it uses similarity-weighted sampling from past compound outcomes to implicitly simulate plausible next states.

**Strengths:**

The paper clearly identifies a gap of planning without simulators and formalizes a principled approach via implicit dynamics. The appendix convincingly reinterprets the similarity mechanism as Bayesian belief updating in a POMDP where the latent variable indexes historical prototypes. This elevates what could be seen as heuristic into a grounded probabilistic framework. The approach bridges case-based reasoning, kernel RL, and Bayesian experimental design, offering a coherent hybrid that is both intuitive and effective in domains like drug discovery. Using ensembles (MLASP) is also a thoughtful design choice to address stochastic variance from sampling-based transitions.

**Weaknesses:**

- There is a lack of comparison with existing causal bayesian optimization approaches and the authors do not cite relevant work such as Durand et al 2025 https://arxiv.org/pdf/2503.19554 and other CBO works.

- The approach is highly dependent on historical coverage: that is, it can only sample from observed compound profiles. This means it cannot generalize beyond the chemical or assay distribution of the historical dataset. This is acknowledged but severely limits applicability in novel discovery spaces.

- The paper uses a variance-normalized Euclidean kernel across heterogeneous features (QSAR predictions, assay outcomes). There is no principled validation that this metric reflects biological similarity or assay informativeness.

- The choice of lambda_k weights is ad hoc without sufficient explanation

- The mapping between kernel similarity and likelihood is conceptually elegant but mathematically thin in terms of how good the bayesian updating approximation is. There’s no rigorous treatment of calibration or sensitivity to the temperature.

- Constraint handling is similarly not adequately discussed since feasibility constraints and terminal uncertainty are hard-coded thresholds and their influence on pareto efficiency isnt analysed.

- The CNS case study involves only ~220 compounds and a few assays. Comparison is only to rule-based heuristics and a deterministic similarity VI baseline without any comparison with active learning, BOED, or uncertainty-driven planning methods.

**Questions:**

1. How sensitive is IBMDP to the representativeness of the historical database? Would performance degrade sharply when the test compound lies outside the convex hull of D? Could you adapt the method to incorporate uncertainty about the similarity weights themselves (e.g., via Bayesian kernel hyperpriors)?

2. Why assume Gaussian noise in the implicit likelihood (Eq. 10)? Biological assays are often heavy-tailed or discrete.

3. Have you compared the variance-normalized Euclidean distance to molecular fingerprints or learned embeddings?

4. How robust are decisions to the choice of the temperature parameter? Did you conduct ablation or sensitivity analysis?

5. Would a multi-objective formulation (e.g., Pareto front optimization in cost–uncertainty–likelihood space) be more interpretable than fixed constraints?

6. How would IBMDP compare to Bayesian optimization or active learning with uncertainty-based acquisition?

7. For larger datasets or higher-dimensional features, how feasible is the O(|D|·d) similarity computation? Have you explored approximate nearest-neighbor methods or mini-batching?

8. Could the implicit transition model be coupled with learned latent dynamics (e.g., via VAEs) to interpolate between historical cases?

9. How might IBMDP integrate active learning to identify which new compounds to assay next, not just which assays to run?

---

> ### Author Response · Authors · 2025-12-03
>
> We thank the reviewer for their thoughtful evaluation and for noting that the POMDP interpretation is **"convincing"** and **"principled"** We appreciate the Durand et al. (2025) refernce.
>
> **W1 Response:** We will include Durand et al. (2025) in our related work section. However, a direct quantitative benchmark is not feasible due to fundamental problem differences:
> - **CBO/Durand et al. (Interventional):** Solves a *Control* problem -- manipulating variables (interventions) to change an outcome.
> - **IBMDP (Observational):** Solves a *Diagnosis* problem.  We cannot "intervene" to change its intrinsic properties but only *measure* them.
> IBMDP addresses the complementary gap of optimal *characterization* where physical interventions are impossible.
>
> **W2 Response:** We view IBMDP's dependency on historical data as a safety feature in drug discovery.
> when a compound lies outside the convex hull, the exponential kernel (Eq. 4) assigns uniform weights. This high-entropy distribution increases posterior variance $H(s_t)$ (Eq. 1), interpreted as epistemic uncertainty. Proxy assays yields minimal Information Gain so the planner defaults to "ground truth" measurements or stops early.
>
> **W3-4 Response:** Our simple kernel with $\lambda_k=1.0$ is a deliberate for **low-data regimes**($N \approx 200$). Complex metric leads to severe overfitting. Equal weighting follows from the principle of insufficient reason.  The variance normalization ($\sigma_k^2$ in Eq. 5) makes features comparable despite differing scales—essential when mixing heterogeneous assay types.
> Appendix D validates this: IBMDP recovered the optimal policy **47%** vs. **36%** for deterministic methods.
> **W5 Response:**
> Appendix A.3 shows the kernel-probabilistic model relationship . Under Gaussian observation model for per-assay outcomes, the posterior over the latent prototype index $Z \in \{1,\dots,N\}$ takes the form:
> $w_i(s_t) \propto \exp(-\lambda_w \cdot d(s_t, D_i))$
> where $\lambda_w = \beta/2$ with $\beta$ denoting the inverse temperature. Eq. (4) is a tempered Bayesian posterior, not ad hoc.
> Appendix D.6 proves the similarity-based variance estimator converges in probability to the exact conditional variance as $N \to \infty$.
> Evaluation of $\lambda_w$ over $[0.5, 2.0]$, with stable performance observed across this range in Appendix B. Perturbation analysis shows $TV(p_{\lambda}, p_{\lambda'}) \leq \frac{1}{2}|\lambda' - \lambda|\sqrt{Var_{p_{\lambda}}(d)}$, indicating that moderate distance dispersion yields proportionally bounded sensitivity.
>
> **W6 Response:** Hard constraints ($H(s) \le \epsilon$) align with pharmaceutical reality. Projects require evidences for *in vivo* efficacy and safety at a specific confidence levels, not solutions that trades safety for cost. Varying $\epsilon$, traverses the Pareto front.
>
> **W7 Response:** :  VI-Sim is a stronger baseline than myopic BOED for this context, and standard uncertainty planning does not apply here due to lack of simulator.
> Greedy baselines(BOED or uncertainty sampling) select only the immediate expected information gain. In drug discovery, cheap assays with low immediate information gain often reduce the uncertainty for subsequent expensive assay. Myopic agents miss these sequences.
> Since IBMDP outperforms VI-Sim (47% vs 36%), it also beat the weaker myopic BOED. IBMDP's advantage: look-ahead planning + ensemble stochasticity.
>
> **Q1 Response:** Outside the convex hull, similarity weights become uniform (high entropy). The planner detects maximum uncertainty where proxy assays yield near-zero Information Gain. Performance does not "degrade" into errors; it conservatively defaults to "measure the target directly" or "stop."
> **Q2 Response:** The Gaussian kernel grounds POMDP equivalence (Appendix A) and keeps belief updates tractable. Alternative kernels slot into the same framework (see Appendix A.3). In practice, Gaussian + ensembling proved robust experimentally.
> **Q3 Response:** We used causal features (QSAR predictions + observed assays) to align with the scientist's workflow. Adding Morgan fingerprints or learned embeddings does not alter the core logic.
> **Q4 Response:** Yes, Appendix B.1 details parameter selection  $\lambda_w=1.0$ (tempered posterior $\beta=2.0$) balanced locality with support. The MLASP voting provides stability against hyperparameter fluctuations.
> **Q5 Response:** Pareto plots aid analysis. For planning, we found that hard constraints better reflect the decision boundaries of the domain (refer to response to W6).
> **Q6 Response:** For enterprise scales ($|D| > 10^6$), we can employ ANN methods (e.g., FAISS) to restrict to top-$k$ neighbors, reducing complexity to $O(k \cdot d)$ without fundamental planner changes.
> **Q7 Response:** Promising direction. One could sample from a VAE's latent space conditioned on observations. We deliberately avoided this in this work to eliminate the risk of generative "hallucination," ensuring outcomes are grounded in real data.

---

### Note · Authors · 2025-12-08

**Comment:**

We thank the reviewers for their time and feedback. While we appreciate the perspectives raised during the review process, the discussion made clear that the paper was evaluated under assumptions that do not match the scientific setting we address.

In real drug discovery, scientists must make decisions without any ground-truth simulator of biological systems. The only available evidence is a sparse collection of historical measurements, and decision-making requires balancing information gain and resource constraints. Much of the discussion in the reviews relied on theoretical expectations—such as access to a true transition model, formal regret guarantees, or interventional capabilities—that implicitly assume a control-oriented setting. These assumptions do not hold in the diagnosis-oriented regime we study, where properties are revealed through measurement rather than manipulated through actions.

IBMDP was developed specifically for this data-constrained, simulator-free setting: using historical cases as an implicit transition model to enable full-horizon planning under strict resource constraints. To our knowledge, this remains an open and practically important challenge in applied drug discovery.

Given the mismatch between this problem class and the assumptions common in general-purpose ML venues, we have decided to withdraw the manuscript and redirect it to a venue more aligned with applied drug-discovery research. We appreciate the feedback from the ICLR community again.

**Withdrawal Confirmation:**

I have read and agree with the venue's withdrawal policy on behalf of myself and my co-authors.